## Registered report

cognition/psychology

metacognition, Dunning–Kruger effect, overconfidence

**Author for correspondence:**
Robert D. McIntosh
e-mail: r.d.mcintosh@ed.ac.uk

# Skill and self-knowledge: empirical refutation of the dual-burden account of the Dunning–Kruger effect

Robert D. McIntosh, Adam B. Moore, Yuxin Liu and Sergio Della Sala

Human Cognitive Neuroscience, Psychology, University of Edinburgh, Edinburgh, UK

 RDM, 0000-0002-7615-6699; ABM, 0000-0002-7002-5460;
YL, 0000-0001-9034-0030; SDS, 0000-0003-4719-508X

For many intellectual tasks, the people with the least skill overestimate themselves the most, a pattern popularly known as the Dunning–Kruger effect (DKE). The dominant account of this effect depends on the idea that assessing the quality of one's performance (metacognition) requires the same mental resources as task performance itself (cognition). Unskilled people are said to suffer a *dual burden*: they lack the cognitive resources to perform well, and this deprives them of metacognitive insight into their failings. In this *Registered Report*, we applied recently developed methods for the measurement of metacognition to a matrix reasoning task, to test the dual-burden account. Metacognitive sensitivity (information exploited by metacognition) tracked performance closely, so less information was exploited by the metacognitive judgements of poor performers; but metacognitive efficiency (quality of metacognitive processing itself) was unrelated to performance. Metacognitive bias (overall tendency towards high or low confidence) was positively associated with performance, so poor performers were appropriately less confident—not more confident—than good performers. Crucially, these metacognitive factors did not cause the DKE pattern, which was driven overwhelmingly by performance scores. These results refute the dual-burden account and suggest that the classic DKE is a statistical regression artefact that tells us nothing much about metacognition.

## 1. Introduction

### 1.1. Skill and self-knowledge

The arrogant blowhard, blind to his own incompetence, is a familiar stereotype throughout human history. This salient aberration of

self-knowledge has been lamented by eminent thinkers, from Confucius[1] to Darwin.[2] More recently, it has surfaced within psychological science as a marker of the Dunning–Kruger effect (DKE): those with the least skill in a given domain overestimate themselves the most. This pattern is found for diverse intellectual tasks, regardless of whether people rate themselves in absolute terms or relative to their peers (see [1], for a review). For instance, following a test of logical reasoning, participants in the bottom quartile, performing around the 12th percentile, ranked themselves on average at the 68th percentile; by contrast, top quartile participants underestimated their true standing, though less severely [2]. In popular culture, the DKE has morphed into the modern trope that *stupid people are too stupid to know they are stupid*.[3]

This trope simplifies, and yet captures the essence of Kruger & Dunning's [2] account of their data. They proposed that, in many intellectual tasks, assessing the quality of one's performance (metacognition) requires the same mental resources as task performance itself (cognition). An example would be that the knowledge needed to judge the grammaticality of a sentence is exactly that which would be needed to compose similar sentences. Given this proposed overlap of cognitive and metacognitive resources, unskilled people are said to suffer a *dual burden*: they lack the cognitive resources to perform well, and this simultaneously deprives them of insight into their failings, so that they think themselves to be more competent than they really are. Conversely, skilled people have the resources both to perform well and to evaluate their performance. According to the dual-burden account, these variations in task performance and metacognition, both linked to underlying task skill, drive the negative correlation between task skill and estimation error that defines the DKE.

However, the explanatory role of metacognitive insight has been subject to scepticism [3–7]. One reason for this is that typical demonstrations of the DKE are prone to statistical artefacts, which might provide a simpler account of the pattern. The most obvious artefact is regression to the mean, which can arise if the objective measure of performance used to rank participant skill also enters into the calculation of estimation error, as in the original studies of Kruger & Dunning [2]. Random factors that bias the measure of performance (up or down) will tend to bias estimation error in the opposite direction when subtracted from the self-estimate, promoting overestimation among the poorest performers, and underestimation among the best. Regression to the mean certainly contributes to the classic DKE, because the DKE is reduced in strength when steps are taken to control for this artefact [4–6,8–11]. The most effective control is to use independent subsets of trials to index performance and to calculate estimation error.

But, even if regression to the mean is controlled for, the DKE could still be an artefact, related to limitations of the typical methods of studying self-estimation, by asking participants for global estimates of their percentile rank or absolute score. An individual who reports their performance accurately might indeed have good insight, or they might just have made a lucky guess. Smaller estimation errors do not necessarily imply greater insight, and one-shot global reports cannot provide adequate measures of metacognition [12]. On this basis, several authors have argued that the DKE implies nothing in particular about metacognitive differences between skilled and unskilled people, but that people *in general* are poor at giving global estimates of their ability [4,6]. Such estimates are imprecise (noisy), and tend to regress towards some common default. If that default guess is optimistic, and there is much evidence that people *in general* rate themselves above average [13–15], then it will be a gross overestimate for an unskilled person, and closer to the truth for a more-skilled person. Under this *noise-plus-bias* account, the DKE is attributable to uncertain self-estimation in the context of performance differences between more- and less-skilled people, and a general tendency for people to rate themselves as better than average.

Meaningful measures of metacognition require a more detailed, psychophysical approach, relating trial-by-trial variations in self-estimation to objective performance [12]. We implemented such an analysis for simple movement and memory tasks: pointing at a dot or recalling its position after a delay [11]. To eliminate regression to the mean, we measured task skill during a preliminary block of trials, independent of our measures of task performance and metacognition. First, we replicated the classic DKE pattern, showing that it generalizes beyond the intellectual domain to these more basic movement and memory tasks. Second, our psychophysical assessment of metacognition suggested that poor performers did indeed have poorer metacognitive insight, consistent with the dual-burden account. Third, we used a path analysis approach to model the dual-burden account, assessing the extent to

---

[1]'Real knowledge is to know the extent of one's ignorance.'

[2]'Ignorance more frequently begets confidence than does knowledge.'

[3]See, for example, https://www.youtube.com/watch?v=wvVPdyYeaQU.

which the negative relationship between task skill and estimation error (i.e. the DKE pattern) was driven by variations in task performance and metacognitive insight. For movement and memory tasks alike, we found only a weak, non-significant net contribution of metacognitive factors, with the DKE driven instead by task performance. Less-skilled participants performed more poorly *and* had poor metacognitive insight, but it was the poor performance and not the lack of insight that caused their apparent overestimation.

This implies that the dual-burden account of the DKE may be correct in its general premise that unskilled people lack metacognitive insight, but mistaken in its assumption that this causes their propensity for inflated self-estimation. However, these conclusions were based on an analysis of movement and memory tasks, so cannot necessarily be generalized to the higher-level intellectual tasks, such as logical reasoning, for which the DKE was originally described [2]. A proper test of the dual-burden account requires an analysis of metacognition in a higher-level task more representative of this literature. In proposing such an analysis, we can take advantage of a powerful theoretical framework developed recently for the modelling and measurement of metacognition.

## 1.2. Measuring metacognition

Fleming & Daw [16] have argued that metacognition is best explained via a second-order computational model, in which metacognitive processing is coupled with, but distinct from, first-order processes underpinning cognitive performance. They apply methods developed by Maniscalco & Lau [17,18], for measuring the information accessed by metacognition (metacognitive sensitivity), and the quality of metacognitive processing itself (metacognitive efficiency). The general analysis strategy is an extension of classical signal detection theory [19].

Consider first a simple cognitive (perceptual) task in which a participant must discriminate between stimulus classes. Each is assumed to generate neural evidence, normally distributed with equal variance, along an internal decision axis. The participant's sensitivity to the difference between stimulus classes is defined by the distance between the two distributions (in units of standard deviation); but the discrimination responses they make are also shaped by their criterion or bias (i.e. where on the decision axis they set their criterion for switching the response between stimulus classes). Signal detection theory provides an elegant set of methods to characterize sensitivity ($d'$) and bias separately. In this context, $d'$ estimates the participant's sensitivity for the first-order cognitive discrimination.

But if we are also interested in metacognition, then we can ask the participant to make an additional response on each trial, to rate how confident they are in their cognitive judgement. This (second-order) response is a *metacognitive* discrimination: the participant is reporting on the quality of their own cognitive processing. These confidence ratings can be used to estimate *metacognitive* sensitivity. Although computationally complex, the analysis is reasonably straightforward in conceptual terms [12,16–18]. If the participant is sensitive to the quality of the information on which their cognitive responses are based, then they should give higher confidence ratings for accurate than for inaccurate responses. To estimate metacognitive sensitivity, we pragmatically assume that the participant is a metacognitively ideal observer, whose confidence ratings are made with perfect insight into the information on which their cognitive responses are based, and we estimate what their first-order $d'$ would be if that were true. This value, known as *meta-d'*, or metacognitive sensitivity, indexes the information accessed by metacognition, expressed in the same units as first-order $d'$.

If the participant really is a metacognitively ideal observer, then *meta-d'* should be equal to $d'$, because metacognitive and cognitive responses will be based on identical information. Typically, *meta-d'* may be lower than $d'$, because it is based on a subset of the first-order information, or on a noisy or decayed transformation of it [17].[4] The extent to which *meta-d'* approaches the ideal value set by $d'$ indexes how fully the first-order information has been accessed by metacognitive processes, thus how informationally efficient metacognitive processing is. This furnishes a second measure of metacognition, *metacognitive efficiency*, usually given by the proportional measure *meta-d'/d'*. While metacognitive sensitivity indexes the evidence accessed by metacognitive processes, and is limited by first-order cognitive sensitivity, metacognitive efficiency indexes the quality of metacognitive processing itself. Metacognitive sensitivity would generally be expected to track cognitive performance more or less closely, but it is an open question how metacognitive efficiency will vary with cognitive ability for any given task.

---

[4]However, a second-order metacognitive model also allows that, under some circumstances, *meta-d'* may be higher than $d'$, if metacognition can access additional information not available to cognition, or is influenced by higher-level beliefs about the likelihood of success or failure (see [16]).

**Table 1.** Summary of dependent measures for investigating skill and self-knowledge in a matrix reasoning task. (*a*) Dependent measures typical of the standard framework within which the DKE has been studied and (*b*) further cognitive and metacognitive measures to be used in the present study.

| dependent measure | block | conceptual role of measure | method of calculation |
|---|---|---|---|
| *(a) standard DKE measures* | | | |
| cognitive skill | baseline | index of task ability | percentage correct in baseline block of matrix reasoning task (with performance between chance and ceiling remapped to a 0–100 scale) |
| cognitive performance | test | index of task performance | percentage correct in test block of matrix reasoning task (with performance between chance and ceiling remapped to a 0–100 scale) |
| relative estimation error | test | over- or under-estimation of own rank position relative to others | actual percentile rank subtracted from estimated percentile rank |
| absolute estimation error | test | over- or under-estimation of own performance in absolute terms | actual cognitive performance score subtracted from estimated score |
| *(b) further cognitive and metacognitive measures* | | | |
| cognitive sensitivity | test | information content of cognitive responses in matrix reasoning task | $d'$ from signal detection theoretic analysis of cognitive responses |
| metacognitive sensitivity | test | information exploited by metacognitive confidence judgements | *meta-$d'$* from signal detection theoretic analysis of metacognitive ratings |
| metacognitive efficiency | test | quality of metacognitive processing | *meta-$d'$* expressed as a proportion of $d'$, indexing the proportion of the information content of cognition that is exploited by metacognition |
| metacognitive bias | test | overall tendency towards high or low confidence, independent of performance | unweighted mean confidence rating, across correct and incorrect responses |

One much simpler metacognitive measure may also be relevant to consider. When metacognition is investigated via confidence ratings, then the (unweighted) average confidence across correct and incorrect judgements provides an index of the person's overall bias towards high or low confidence, independent of first-order response accuracy (e.g. [12,17,18,20]). We can potentially use this *metacognitive bias* measure to investigate differences between individuals in the use of the confidence scale. Table 1 gives an overview of the dependent measures for this study, including the cognitive and metacognitive variables reviewed in this section.

## 1.3. Metacognition in the Dunning–Kruger effect

The dual-burden account proposes that the DKE arises specifically for tasks in which metacognitive insight depends on *exactly the same* information as first-order cognition [1,2]. In Fleming and Daw's terminology, this would be a first-order model of metacognition, which is a special case of the more general second-order model. For this special case, metacognitive performance is based on the same information as cognitive performance, so cognition and metacognition would be tightly coupled. Empirically, this overlap of information for cognition and metacognition predicts a strong positive relationship between cognitive sensitivity ($d'$) and metacognitive sensitivity (*meta-$d'$*).

It should be noted that, within the metacognitive framework discussed, the same positive relationship would be expected simply from the fact that metacognitive sensitivity is constrained by the information available to cognition. As such, a positive relationship between these variables is rather likely, but relatively uninteresting in psychological terms, because differences in metacognitive sensitivity

dependent on task performance do not in themselves imply any differences in the *quality* of metacognitive processing between people with high and low ability for a task. However, as noted by Fleming & Lau [12], it is also possible that there are real differences in the quality of metacognitive processing between high- and low-skill participants, which would be reflected in the measure of *metacognitive efficiency*. If participants of lower ability were found to have lower metacognitive efficiency, rather than just lower metacognitive sensitivity, this would support a psychologically stronger, more interesting reading of the dual-burden hypothesis.

Finally, we can potentially use the *metacognitive bias* variable to investigate the idea that poor performers are overconfident about their cognitive judgements. This is not a critical part of the dual-burden account, but it is a characterization of poor performers that is prevalent within the same literature (e.g. [1,2,5,21,22]). A negative relationship between cognitive sensitivity and metacognitive bias would support this characterization.

## 1.4. The present study

In this study, we apply a computational model of metacognition to a matrix reasoning task, epitomizing the kind of intellectual skill for which the DKE was first described [2]. The design will allow us to more fully evaluate the assumptions of the dual-burden account, in terms of the relationship between cognitive ability and different dimensions of metacognitive insight, and to quantify the relative influences of cognitive and metacognitive factors in driving the famous DKE pattern.

# 2. Methods

## 2.1. Participants

Participants were current university students or alumni, aged from 18 to 50, otherwise unselected for age, sex, nationality or academic discipline.[5] We had a pre-registered target of 150 valid datasets, as guided by *a priori* power considerations (see §2.6). In practice, we recruited 159 participants and obtained 151 valid datasets (see §3.1).

## 2.2. Stimuli

The primary task was a matrix reasoning task, adapted from an open-access matrix reasoning item bank [23]. Each puzzle was based on a $3 \times 3$ matrix of shapes, with the lower right element missing, and the participant was required to identify the missing element from among candidate solutions. To do this successfully, participants must deduce relationships between the shapes of the matrix, which requires them to take account of one or more stimulus dimensions (shape, colour, size, relative position). In the original version of this task, four candidate solutions were presented per puzzle, with a time limit of 30 s to respond (for details, see [23]). In our adapted version, only two candidate solutions were presented, one correct and one incorrect, so that each puzzle was a two-alternative forced-choice, suitable for a signal detection theoretic analysis (figure 1).

The open-access matrix reasoning item bank (https://osf.io/g96f4/) comprises three sets of 80 puzzles. These three puzzle sets are exactly parallel, but differ in the particular shapes used. To adapt these materials for our purposes, we first characterized the properties in the puzzles in one parallel set (set 1), coding the number of stimulus dimensions required to solve each puzzle, and rating the difficulty of each on a 1–4 scale.[6] Thirty-one puzzles were classed as both two-dimensional and of intermediate difficulty (rating 2), and we selected the first 30 of these to define a basic puzzle set for our task. We then designated 10 of the puzzles from this basic set of 30, specifically every third puzzle (3, 6, 9,…, 30), as baseline puzzles, with the other 20 being designated as test puzzles. This distinction relates to the role that these puzzle items will play in the analysis of the DKE. Baseline

---

[5]The original protocol stated that all participants would be current students of the University of Edinburgh, aged 18–40. The difficulty of recruitment during the COVID-19 pandemic meant
that, in order to achieve sufficient numbers, we relaxed these criteria slightly, recruiting current and previous students of any university, below the age of 50. This protocol deviation was approved retrospectively on 25 July 2022.

[6]An objective index of difficulty was publicly available for a group of 106 adults performing the original task [23], although only 38 of the puzzles had been completed by 20 or more of these participants (range 20–106). For these 38 puzzles, the observed mean accuracy correlated with our subjective difficulty rating at $r = -0.77$, giving us confidence in the validity of our difficulty ratings for the full set.

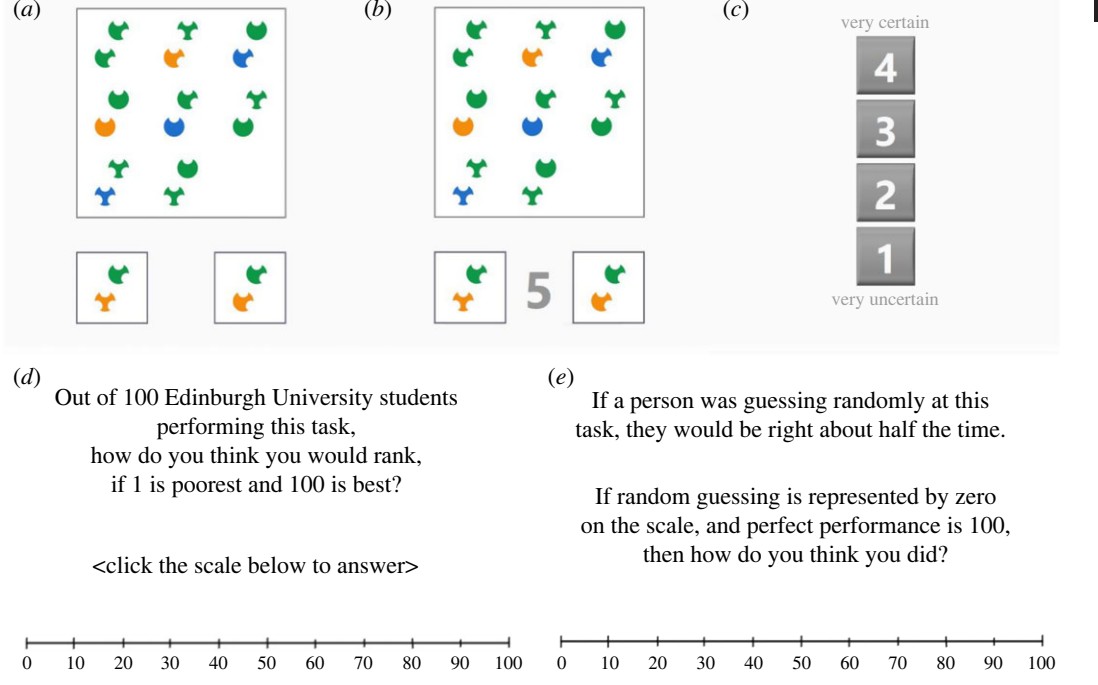

**Figure 1.** Trial events: (*a*) a 3 × 3 matrix is presented, with one element missing, and two candidate solutions, and the participant must click on the correct solution; (*b*) a digit countdown appears in the last 5 s of the 17 s display period; (*c*) the first-order response is followed by a confidence rating; (*d*) following the main task, participants make a global self-estimate of relative performance and (*e*) participants then make a global self-estimate of absolute performance.

puzzles will be used to characterize cognitive skill, while test puzzles will be used for the separate calculation of task performance, estimation error and metacognitive measures, allowing us to minimize regression to the mean (see §1.1 for discussion of this artefact).

Next, in order to boost the number of items, we transformed selected puzzles from the original set, to produce novel puzzles, which preserved the essential logical demands of the original (see electronic supplementary material, section S2 for details). We transformed six of the baseline puzzles, and 10 of the test puzzles in this way, bringing the number of baseline puzzles up to 16, and the number of test puzzles up to 30, for puzzle set 1. The equivalent puzzles (and transformations) were then added from puzzle sets 2 and 3, to make a grand total of 48 baseline puzzles and 90 test puzzles, relatively homogeneous for dimensionality and difficulty.

## 2.3 Procedure

Testing took place in a quiet, private room; all data were collected under an anonymous study code, and the participant's responses were not directly observed. As testing took place between 28 January 2022 and 17 August 2022, during the COVID-19 pandemic, the experimenter and participants (except where exempt) wore facemasks for respiratory hygiene.[7]

The trial events are illustrated in figure 1. The participant sees a 3 × 3 matrix of shapes, with the lower right element missing, and two candidate solutions, one correct and one incorrect, below the matrix. The participant must click on the solution that they think is correct. The matrix remains on-screen until a response is made, or for a maximum of 17 s, with a digit countdown visible for the last 5 s. If the time limit is reached, the matrix disappears, and only the candidate solutions remain on screen until the participant responds. In the baseline (and familiarization) phase, the next trial begins after a 1 s delay. In the test phase, before proceeding to the next trial, the participant is required to rate their confidence in the preceding response, by clicking one of a column of four numbered buttons, where 1 is labelled 'very uncertain' and 4 'very certain'.

[7]The use of facemasks was not part of the original Stage 1 protocol, which received in-principle acceptance prior to the COVID-19 pandemic. This protocol amendment was approved retrospectively on 25 July 2022.

Each participant first completes a short familiarization phase of six very easy puzzles, which are not drawn from the experimental set, followed by a baseline phase of 48 trials, divided into three blocks of 16 trials, and then a test phase of 90 trials (with confidence ratings), divided into five blocks of 18 trials. Between each block, a dialogue box pops up with a 'continue' button, allowing the participant to pause until they are ready. An instruction screen before the practice and baselines phases informs the participant that, *You will be shown a series of puzzles with one piece missing, and two possible solutions. Please click on the solution that correctly completes the puzzle. If you take longer than 17 s, the puzzle will disappear, but the solutions will be shown until you choose one of them. You must choose a solution for every puzzle, even if you feel that you are just guessing.* An instruction screen before the first block of the test phase informs the participant that, *You are now starting the main phase of the experiment. You will be shown a series of puzzles as before, but after every puzzle you will also be asked to rate your confidence that the solution you chose was correct. The rating scale goes from 1 (very uncertain) to 4 (very certain). Take time to consider your rating, and please be willing to use the full range of the scale.*

Following the final block of test trials, participants see an information screen which states, *Well done! You have completed the main phase of the experiment. You will now be asked two questions about your performance during the main phase only. Please take time to understand the question and consider your answer. Try to answer as accurately as you can. Click here to continue.* Participants then complete two global self-estimates. The first is a relative estimate prompted by the text, *Out of 100 Edinburgh University students performing this task, how do you think you would rank, if 1 is poorest and 100 is best?* The second is an absolute estimate, prompted by the text: *If a person was guessing randomly at this task, they would be right about half the time. If random guessing is represented by zero on the scale, and perfect performance is 100, then how do you think you did?* Each prompt is shown above a horizontal scale from 0 to 100 (although the scale for the relative response will not register any estimate lower than 1). When the participant clicks on the scale, a line is shown to mark the estimate and a SUBMIT button appears below the scale. The participant can revise their estimate by re-clicking the scale, or can click the SUBMIT button to confirm.

Finally, in a short verbal debrief, the experimenter asks the participant, first, what they think the experiment is studying; and, second, if they have heard of the DKE. If the participant answers positively to the second question, they are asked to give a brief description of this effect.

## 2.4 Dependent measures

### 2.4.1. Standard Dunning–Kruger effect measures

We first extracted standard measures of cognitive skill and errors of global self-estimation, to support a typical analysis of the DKE (table 1*a*).

*Cognitive skill* is indexed by accuracy of matrix reasoning responses in the baseline phase. The raw per cent correct was expected to range from 50 to 100 (i.e. from chance level to perfect performance). We remapped this range to a 0–100 scale by subtracting the chance level (50) and multiplying by two.

*Cognitive performance* is indexed by accuracy of matrix reasoning responses in the test phase, similarly remapped to a 0–100 scale between chance level and perfect performance.

*Relative estimation error.* For each participant, we converted the performance score into a percentile rank and subtracted it from the relative self-estimate made at the end of the test phase, to give relative estimation error. Positive values represent overestimation and negative values underestimation.

*Absolute estimation error.* For each participant, we subtracted the cognitive performance score from the absolute self-estimate made at the end of the test phase, to give absolute estimation error. Positive values represent overestimation and negative values underestimation.

### 2.4.2. Further cognitive and metacognitive measures

To support the psychophysical analysis of metacognition, we also calculated the following cognitive and metacognitive measures from the test phase (table 1*b*).

*Cognitive sensitivity* ($d'$) for each participant was estimated from response data as follows:

$$d' = z(H) - z(FA).$$

Here, $z()$ indicates the inverse of the standard cumulative normal distribution function, $H$ refers to type 1 hit rate and FA is the type 1 false alarm rate. This analysis was performed within the HMeta-d toolbox for Matlab [20].[8]

---

[8]https://github.com/smfleming/HMeta-d

*Metacognitive sensitivity* (*meta-d′*) was estimated within the same toolbox, via Bayesian estimation at the participant level. As described by Fleming [20, pp. 2–7 and Appendix], the likelihood of observed confidence rating data are modelled via a multinomial probability function,

$$L(\theta|\text{data}) \propto \prod_{x,i,j} P_\theta(\text{conf} = x|\text{stim} = i, \text{res} = j)^{n_{\text{data}}(\text{conf}=x|\text{stim}=i,\text{resp}=j)}.$$

Bayesian inference provides a posterior estimate of the parameter values for metacognitive sensitivity (*meta-d′*), and has several advantages over traditional methods, including, but not limited to, lack of need for edge correction, automatic precision for estimates relative to amount of data available, and quantification of uncertainty around estimates (see [24]). For single-participant estimation we used non-response conditional estimation via Markov chain Monte Carlo simulation executed in JAGS.[9] Three chains of 10 000 samples each, with 1000 sample burn-in, were used for convergence. The default prior on *meta-d′* was a normal distribution with mean equal to the observed *d′* computed from the participant's response data, and variance = 2.

*Metacognitive efficiency*, *meta-d′/d′*, was estimated within the same toolbox, as the mean of the ratio of posterior samples of *meta-d′* and *d′*.

*Metacognitive bias* was calculated as the (unweighted) mean confidence rating for correct and incorrect responses. That is, mean confidence was calculated for correct responses, and for incorrect responses separately, and metacognitive bias was the mean of these two values.

## 2.5. Participant exclusions and approach to outliers

Participants were excluded from the main analysis if they scored at floor or ceiling level on either the baseline or test trials of the matrix reasoning task (i.e. raw accuracy less than or equal to 50%, or equal to 100%) (four exclusions were made on this basis, see §3.1). If any participants at debrief were able to describe, approximately or accurately, the DKE, then they were also excluded (no exclusions resulted from this criterion). Regardless of these exclusions, all participants' scores were included when assigning a percentile rank to performance for the calculation of relative estimation error.

Our main analyses were based upon patterns of inter-correlation among key variables. Our general approach to outliers was to treat them as informative data, and not to exclude them, but to focus on robust measures of association, based on ranked data. However, we screened for univariate outliers on the metacognitive efficiency variable, because this was based on a ratio measure (*meta-d′/d′*), which may yield misleading extreme values, especially when *d′* is low (see electronic supplementary material, section S1.3). Here, we used a relatively conservative exclusion criterion, removing only values more than two interquartile ranges below the first quartile, or above the third quartile.

As a robustness check, all analyses were re-run with no participant or outlier exclusions, and are reported in electronic supplementary material, section S5.

## 2.6. Power and sample size

Our pilot study (electronic supplementary material, section S1) led us to anticipate that the strength of the DKE, as measured by the ranked correlation between cognitive skill and estimation error, would be around −0.66 for relative estimation error and −0.45 for absolute estimation error, with a *weakest* expected strength of around −0.57 for relative estimation error and −0.38 for absolute estimation error (see electronic supplementary material, section S1.4). To confirm the existence of the DKE for relative and absolute estimation errors, we targeted the minimum expected effect size (−0.38), with a conventional $\alpha$ of 0.05 adjusted to 0.025 (two-tailed), to account for the two inferential tests performed. We would need a minimum sample size of 127 participants, to achieve a power for these tests in excess of 0.99 [25,26]. However, we tested participants until we had 150 valid datasets (the final valid sample size was in fact 151), to provide additional precision of estimation for other relationships of theoretical interest.

Beyond the two critical tests of the DKE, our focus was on the estimation of these other relationships of interest, and their relative contributions to the DKE. Precision of estimation depends on the empirical sample (since we bootstrapped confidence intervals), but should vary inversely with correlation strength, so that stronger correlations will be estimated more precisely than weaker correlations. For general

[9]http://mcmc-jags.sourceforge.net

guidance, given a sample size of 150, we would expect a prediction interval (two-sided 95% confidence interval) of around 0.31 for a correlation strength of 0.2, e.g. (0.04, 0.35), and around 0.12 for a correlation strength of 0.8, e.g. (0.73, 0.85) [27].

## 2.7. Statistical analysis

Bivariate correlations were performed on ranked data (i.e. Spearman correlations), and reported with bootstrapped 95% confidence intervals. The use of ranked data allowed us to make minimal distributional assumptions, and to take a liberal approach to the retention of outliers (see §2.5). It was also consistent with the common convention in this literature of indexing participant performance by percentile rank (e.g. [2]). We base our main conclusions on interval estimates of ranked correlations (we also report point estimates of Pearson correlation coefficients for comparison).

### 2.7.1. Reliability of cognitive performance

An initial condition for the DKE to emerge is that there should be a clear positive correlation between cognitive skill and cognitive performance (i.e. between baseline and test phases). This is a logical requirement for the concept of repeatable skill to be applied to the matrix reasoning task. Chierchia et al. [23] reported a reliability of $r = 0.71$, across 218 participants, at an average retest delay of 35 days, for parallel forms of the matrix reasoning task from which our stimuli were derived. Our pilot study, in which we used rank-order correlations, suggested a median split-half reliability of $\rho = 0.80$, with 95% of 1000 split-half samples producing a correlation higher than 0.70 (electronic supplementary material, section S1.3). We were thus confident that we would observe good reliability for this task ($\rho > 0.70$). We did not have any minimum requirement for this correlation, but if it were much lower than expected then this would help us to understand any shortfall in, or failure to replicate, the critical DKE pattern.

### 2.7.2. Replication of the Dunning–Kruger effect: relationship between cognitive skill and estimation errors

We could not investigate the underlying basis of the DKE, unless the phenomenon itself was present. This was tested via the ranked correlations between cognitive skill and absolute estimation error and relative estimation error separately. This was the critical outcome-neutral condition for our study to be able to address its aims: *there must be a significant negative correlation between cognitive skill and relative estimation error and/or between cognitive skill and absolute estimation error.* We tested these two correlations, using a two-tailed $\alpha$ of 0.025, as described in §2.6. Further analyses would be unable to support any conclusions about the underlying causes of the DKE, for relative or absolute estimation error, unless the DKE for that type of estimation error was significant.

### 2.7.3. Relationships between cognitive measures and metacognitive measures

We estimated the bivariate relationships between cognitive responses in the baseline and test phases, and three measures of metacognitive insight collected in the test phase: metacognitive sensitivity, metacognitive efficiency and metacognitive bias. This allowed us to more fully explore the proposal of the dual-burden account that metacognitive insight is determined by cognitive ability for a task. We initially estimated the relationships with cognitive sensitivity, as measured in the test phase.

We expected that metacognitive sensitivity would be positively related to cognitive sensitivity. This expectation was motivated *a priori* from the metacognitive model we are using, in which cognitive sensitivity constrains the information potentially available to metacognition (see §1.2). It was also supported by our pilot data, which showed a strong positive relationship between these variables ($r = 0.75$, $\rho = 0.69$; electronic supplementary material, figure S5a). As noted in §1.3, this positive relationship would be consistent with the dual-burden hypothesis, which proposes that metacognitive processing depends on the same resources as cognitive performance. However, under a weak reading of the hypothesis, differences in metacognitive sensitivity could just reflect underlying differences in the cognitive information that metacognition can access, without implying differences in the quality of metacognitive processing itself.

A stronger, more interesting version of the dual-burden hypothesis is also possible, in which the quality of metacognitive processing would itself improve with task ability. This would predict a positive relationship between cognitive sensitivity and metacognitive efficiency (our pilot data hints at such a

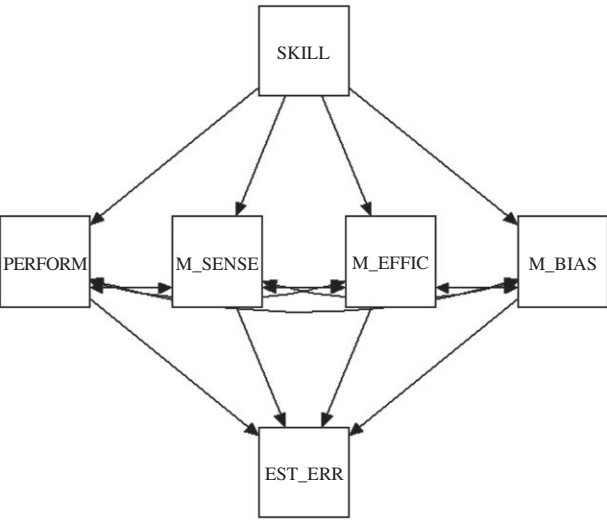

**Figure 2.** *A priori* model for path analysis, with four indirect paths via which the relationship between task skill (SKILL) and estimation error (EST_ERR) could arise: through task performance (PERFORM), metacognitive sensitivity (M_SENSE), metacognitive efficiency (M_EFFIC) and metacognitive bias (M_BIAS). The total role of metacognitive factors in the DKE can be estimated by the summed influence of the three metacognitive paths.

relationship; electronic supplementary material, figure S5a). It should be noted that this would be directionally opposite to the negative association that one might naturally expect when correlating one measure ($d'$) with a fractional index in which it is the denominator (metacognitive efficiency = $meta\text{-}d'/d'$). A positive correlation would thus be much more noteworthy than a negative one.

The relationship of cognitive sensitivity with metacognitive bias may further speak to the widespread idea that poor performers are overconfident. If poor performers truly are overconfident, then they should show a higher metacognitive bias than good performers, inducing a negative relationship with cognitive sensitivity. A bias towards overconfidence among the poorest performers would be consistent with popular characterizations of the DKE, although it is not a specific feature of the dual-burden account.

In addition to estimating metacognitive correlations with cognitive sensitivity in the test phase, we also estimated correlations with cognitive skill, as assessed in the baseline phase. We expected these correlations to be somewhat weaker; but some association between task skill and metacognition would be necessary for metacognitive variables to substantively drive the relationship between task skill and global estimation errors, as the dual-burden hypothesis proposes.

### 2.7.4. Path analysis of the Dunning–Kruger effect

The preceding analyses tested for the presence of the DKE, and examined a general assumption of the dual-burden account, that participants with lower task skill have poorer metacognitive insight. But, even if this assumption is supported, it does not necessarily follow that this plays an important role in shaping the DKE. On the contrary, we previously found that less-skilled participants had poorer metacognitive insight for simple movement and memory tasks, but a path analysis suggested that these metacognitive differences had a negligible influence in driving the DKE pattern [11].

We adopted a similar path analysis here, with robust maximum likelihood estimation [28] to estimate the relative importance of cognitive and metacognitive factors in driving the DKE for the matrix reasoning task. Ranked data were used, and separate path analyses were performed for relative estimation error and absolute estimation error.

The *a priori* model, depicted in figure 2, included four indirect paths via which the relationship between task skill and estimation error (i.e. the DKE) could arise: through cognitive performance, metacognitive sensitivity, metacognitive efficiency and metacognitive bias. We estimated each of these paths, calculating 95% bootstrapped confidence intervals to set plausible ranges on their influence. The total role of metacognitive factors was estimated by the summed influence of the metacognitive paths. The relative strengths of performance and metacognitive paths provided a quantitative estimate of their relative importance in driving the DKE for the present task.

In the case that the confidence intervals on any path included zero, we explored dropping that path and comparing the reduced model(s) against competitor models via the model comparison statistic Akaike information criterion (AIC), seeking a reduced model which represented a better trade-off between variance explained and parameters required. This model selection process was used to determine whether the best model (lowest AIC) included both performance and metacognitive paths, as the dual-burden account requires.

# 3. Results

## 3.1. Sample size, participant exclusions and demographics

In total, 159 complete datasets were collected. One participant was found to have entered the experiment twice (for payment) so her second set of data was removed. Three participants were ineligible for inclusion because they had never been university students. From the remaining sample of 155 participants, four were excluded because of ceiling- or floor-level performance: two performed at floor and one at ceiling in the baseline phase, and one performed at ceiling in the test phase. The four participants excluded on performance criteria were removed after the calculation of performance ranks for the baseline and test phases.

The final valid sample size was 151. By gender, 110 were female, 40 were male and one was non-binary. The mean age was 24.4 years, s.d. 6.2 (median 22, range 18–49). One hundred and twenty-six participants were current university students (120 Edinburgh University), and 25 were alumni (10 Edinburgh University).

## 3.2. Reliability of cognitive performance

An initial pre-condition for the DKE in our analysis is a positive correlation between cognitive skill and cognitive performance (i.e. between baseline and test phases).

The Spearman correlation ($\rho$) was 0.75 (bootstrapped 95% CI (0.66, 0.81)), consistent with our pre-registered expectation of $\rho > 0.70$. The Pearson correlation ($r$) was 0.73 (bootstrapped 95% CI (0.67, 0.80)). The matrix reasoning task thus has very good reliability, providing a sound basis for our evaluation of the DKE in terms of the relation between cognitive skill in the baseline phase and estimation error in the test phase.

The line of best fit between cognitive skill and performance is depicted by the solid line through the filled symbols in figure 3b. This line of fit is above the dotted diagonal of identity, indicating a practice benefit from baseline to test phases. When scores are expressed as percentile ranks, the practice effect is compensated and the solid line of fit approximates the line of identity, showing the general preservation of rank order (figure 3a).

## 3.3. Replication of the Dunning–Kruger effect: relationship between cognitive skill and estimation errors

In figure 3a,b, respectively, the relative and absolute self-estimates of performance in the test phase are indicated by crosses, and the dashed line indicates the relationship with cognitive skill (i.e. baseline scores). Our operational definition of the DKE is as a negative relationship between cognitive skill (in the baseline phase) and estimation error (in the test phase). The estimation error is represented by the discrepancy between the solid (performance) and dashed (estimation) fit lines in figure 3a,b.

The estimation errors are plotted directly in figure 3c,d, showing the expected negative relationships with cognitive skill. For relative estimation error, $\rho$ was −0.57 (bootstrapped 95% CI (−0.66, −0.47)), matching our weakest expected strength of −0.57 (see §2.6), and meeting our two-tailed criterion for significance at the 0.025 level ($p = 1.6 \times 10^{-14}$; $r$ was −0.53 (bootstrapped 95% CI (−0.62, −0.43)). For absolute estimation error, $\rho$ was −0.43 (bootstrapped 95% CI (−0.55, −0.28)), close to our expected strength of −0.45 (see §2.6), and meeting our two-tailed criterion for significance at the 0.025 level ($p = 5.0 \times 10^{-8}$); $r$ was −0.42 (bootstrapped 95% CI (−0.54, −0.27)).

The DKE relationship was thus replicated using the standard method of global self-estimation, more strongly for relative than for absolute estimation errors. In both cases, the data satisfy the pre-registered outcome-neutral criterion for our study to be able to address its aim of investigating the role of metacognitive factors in the DKE.

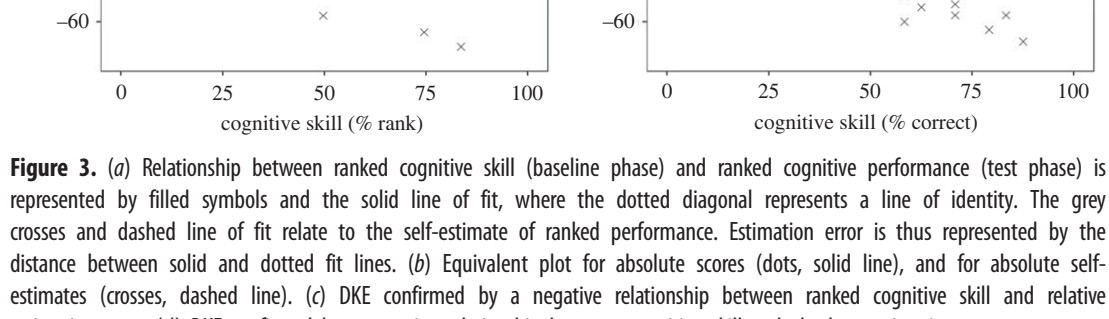

**Figure 3.** (a) Relationship between ranked cognitive skill (baseline phase) and ranked cognitive performance (test phase) is represented by filled symbols and the solid line of fit, where the dotted diagonal represents a line of identity. The grey crosses and dashed line of fit relate to the self-estimate of ranked performance. Estimation error is thus represented by the distance between solid and dotted fit lines. (b) Equivalent plot for absolute scores (dots, solid line), and for absolute self-estimates (crosses, dashed line). (c) DKE confirmed by a negative relationship between ranked cognitive skill and relative estimation error. (d) DKE confirmed by a negative relationship between cognitive skill and absolute estimation error.

## 3.4. Relationships between cognitive measures and metacognitive measures

The next analyses chart the relationships between three measures of metacognition extracted from trial-by-trial confidence ratings in the test phase, and first-order cognitive sensitivity for the same trials. We chose cognitive sensitivity ($d'$) as the relevant measure of performance, because it is on an equivalent scale to the measure of metacognitive sensitivity (*meta-d'*), but is otherwise closely equivalent to the simple % correct measure of cognitive performance ($\rho = 0.99$, $r = 0.96$). The three relationships of interest are plotted in figure 4a–c, and the corresponding correlations are tabulated on the right side of table 2.

As expected, metacognitive sensitivity was strongly positively related to cognitive sensitivity. Poorer performers have lower metacognitive sensitivity in discriminating their successes from their failures, indicating that they have lower quality information available to support metacognitive confidence judgements. The line of best fit approximated the line of unity, indicating that cognitive and metacognitive judgements had access to the same information.

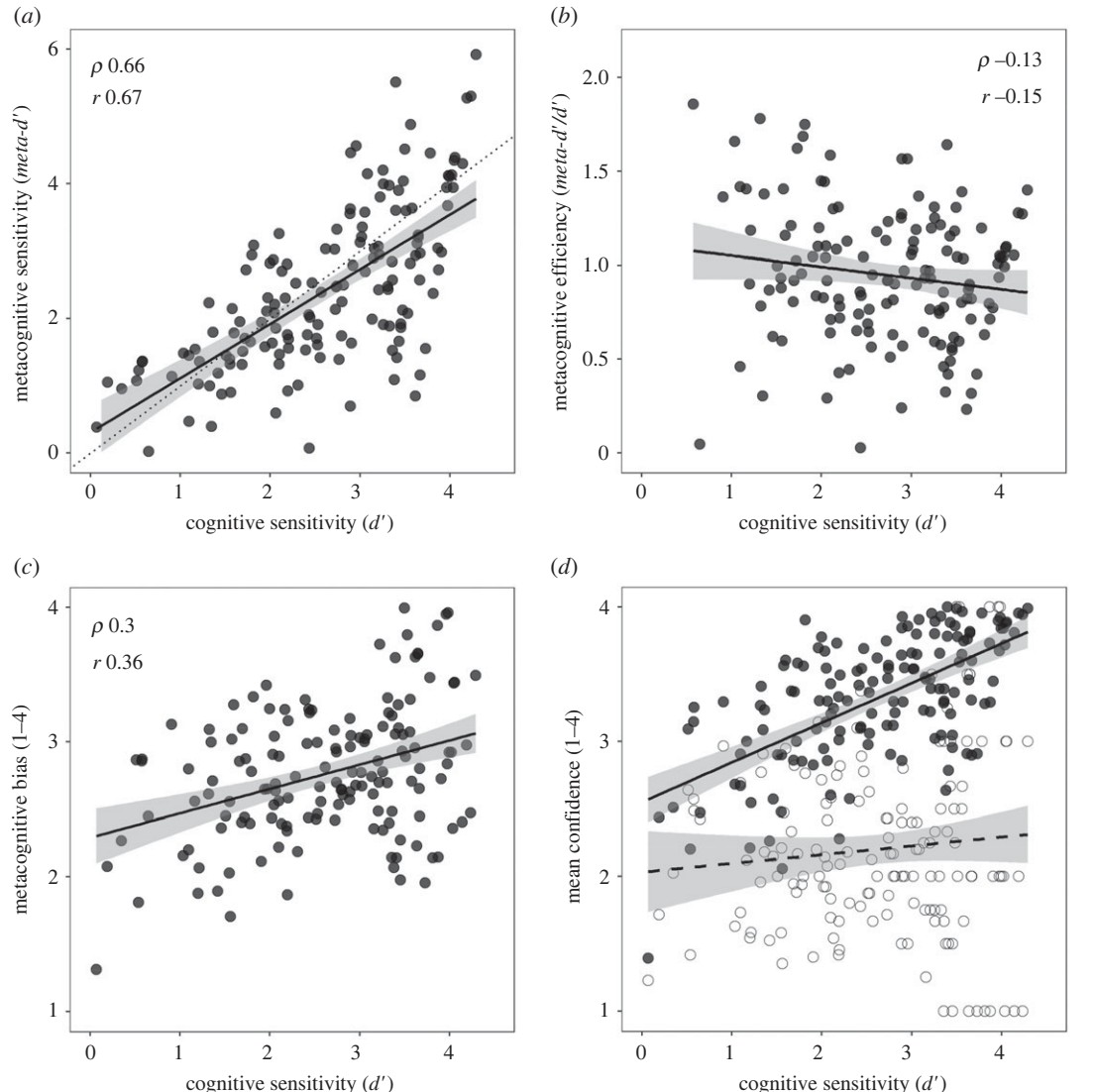

**Figure 4.** Scattergram and line of best fit for relationship between cognitive sensitivity and (*a*) metacognitive sensitivity, (*b*) metacognitive efficiency and (*c*) metacognitive bias. (*d*) The relation between cognitive sensitivity and mean confidence for correct responses (filled symbols, solid fit line) and incorrect responses (open symbols, dashed fit line).

**Table 2.** Correlations (bootstrapped 95% CI) between measures of metacognition in the test phase and cognitive skill (baseline phase) and cognitive sensitivity (test phase). The relationships with cognitive sensitivity are those plotted in figure 4*a–c*.

|  | cognitive skill (baseline phase) | | cognitive sensitivity (test phase) | |
| --- | --- | --- | --- | --- |
|  | Spearman ($\rho$) | Pearson (*r*) | Spearman ($\rho$) | Pearson (*r*) |
| metacognitive sensitivity | 0.50 (0.37, 0.62) | 0.50 (0.38, 0.62) | 0.66 (0.50, 0.80) | 0.67 (0.59, 0.76) |
| metacognitive efficiency | −0.12 (−0.28, 0.05) | −0.13 (−0.28, 0.03) | −0.13 (−0.29, 0.05) | −0.15 (−0.34, 0.06) |
| metacognitive bias | 0.36 (0.21, 0.51) | 0.42 (0.25, 0.55) | 0.30 (0.13, 0.45) | 0.36 (0.20, 0.51) |

For metacognitive efficiency, six high outliers were removed according to the pre-registered exclusion criterion (greater than two interquartile ranges above the third quartile). As anticipated, these outliers all occurred in the lower range of cognitive sensitivity, as a consequence of dividing *meta-d′* by a *d′* value close to zero. For the remaining 145 participants, there was no support for the idea that the quality of

metacognitive processing was worse among poor performers, once differences in the quality of information available to metacognition were accounted for.

Metacognitive bias showed that, contrary to the popular characterization of poor performers as overconfident, they were in fact less confident than their higher-performing counterparts. Figure 4d unpacks the pattern of confidence according to whether the first-order response was correct or incorrect. All people were unconfident to roughly the same degree (less than 2.5) when answering problems incorrectly, but higher performers were much more confident when answering correctly. When the poorest performers answered puzzles correctly, their average confidence rating was only slightly higher than the highest performers when answering incorrectly. Far from being overconfident in their self-assessments, poor performers were appropriately humble (see §4.3).

The above analyses focus on the relationship between metacognition and cognitive sensitivity measured within the same series of trials (test phase). Given that the matrix reasoning task has good test–retest reliability, a broadly similar set of relationships should hold between metacognitive measures in the test phase and cognitive skill (as measured in the baseline block). The left side of table 2 shows that this is true, albeit with a slightly weaker relationship for metacognitive sensitivity. Some level of relation between task skill and metacognition is necessary for it to be possible that metacognitive factors could contribute to the DKE, as the dual-burden account requires.

## 3.5. Path analysis of the Dunning–Kruger effect

To evaluate how well a dual-burden account explains the DKE pattern, we ran a series of path analyses, separately for relative and absolute estimation errors. The *a priori* model, which is shown in figure 2, has four paths that could mediate the overall negative relationship between task skill and estimation error: through cognitive performance, metacognitive sensitivity, metacognitive efficiency and metacognitive bias. The total role of metacognitive factors can be estimated by the summed influence of metacognitive paths. Following our pre-registered plan (§2.7.4), we first fitted the full model, and then tried dropping paths for which the confidence intervals included zero, to potentially identify a reduced model that provides a better account of the data.

Table 3 summarizes the full and the reduced models for relative and absolute estimation error. Full information of all models tested are available in electronic supplementary material, section S3; we focus here on the total path strengths, and the overall quality of the model in terms of adjusted goodness of fit and AIC. When interpreting the path strengths, it should be remembered that the DKE relationship itself is a negative correlation between task skill and estimation error. Therefore, negatively signed paths tend to promote the DKE relationship, and positively signed paths tend to counteract it.

For relative estimation error, the full model provided a good fit to the data ($\chi^2 < 1$, $\chi^2_{\text{robust}} < 1$) and suggested that the overall DKE strength of −0.54 was due to a strong path via cognitive performance (−0.69) tempered by an opposing influence of metacognitive factors (0.15). According to this full model, only cognitive performance drives the DKE, while metacognition (in total) paradoxically tends to counteract it. However, the path via metacognitive efficiency was not significant, and removing it led to a reduced model with a better fit to the data, with significant paths for cognitive performance, metacognitive sensitivity and metacognitive bias (albeit the latter was very weak). This reduced model supported the same conclusion as the full model: performance factors drive the DKE (−0.68), while metacognitive factors have a weaker, directionally opposite influence (0.14).

For absolute estimation error, the full model was a marginal fit to the data ($\chi^2_{(1)} = 3.0$, $p = 0.08$, $\chi^2_{\text{robust}(1)} = 3.61$, $p = 0.06$); the overall DKE relationship was weak (−0.29), and none of the component paths were individually significant. The model reduction process led to the removal of the path due to metacognitive efficiency and then metacognitive bias, leading to a better-fitting reduced model that supported the same general conclusions as for the relative estimation error model above: performance factors dominate the DKE (−0.45), with a weaker, opposing influence attributable to metacognitive sensitivity (0.15).

The reduced models in table 3 clearly contradict the dual-burden account, because while cognitive performance acts to promote the DKE (as expected), the influence of metacognitive differences is directionally opposite. These models are the outcome of a pre-registered model reduction process, which did not allow for the dropping of paths that had a significant influence. However, all of the models were dominated by the path through cognitive performance, with a weaker and less consistent influence of metacognitive factors. If one is committed to a dual-burden account of the DKE, then it makes sense to retain cognitive performance and at least one metacognitive path, but since the observed patterns contradict the dual-burden account, the theoretical rationale to retain any metacognitive path is weakened.

**Table 3.** Path coefficients and 95% confidence intervals for pre-registered (*a priori*) models and best fitting reduced models for both relative and absolute estimation error. AIC, Akaike information criterion. See electronic supplementary material, section S3 for further model comparison statistics, and full information on all models tested. As a further robustness check, we also report the same path analyses for scaled instead of ranked data in electronic supplementary material, section S4.

| | relative estimation error | | absolute estimation error | |
|---|---|---|---|---|
| | *a priori* model | reduced model | *a priori* model | reduced model |
| total DKE | −0.54*** (−0.65, −0.43) | −0.54*** (−0.65, −0.43) | −0.29*** (−0.42, −0.16) | −0.30*** (−0.42, −0.18) |
| performance | −0.69*** (−0.84, −0.55) | −0.68*** (−0.79, −0.57) | −0.17 (−0.39, 0.06) | −0.45*** (−0.57, −0.32) |
| metacognition | 0.15** (0.05, 0.30) | 0.14** (0.08, 0.20) | −0.12 (−0.32, 0.08) | 0.15*** (0.07, 0.23) |
| metacog. sensitivity | 0.11* (0.02, 0.19) | 0.10*** (0.04, 0.15) | −0.11 (−0.25, 0.04) | 0.15*** (0.07, 0.23) |
| metacog. efficiency | 0.00 (−0.02, 0.02) | — | −0.06 (−0.15, 0.03) | — |
| metacog. bias | 0.04* (0.01, 0.08) | 0.04* (0.01, 0.08) | 0.04 (−0.01, 0.09) | — |
| adj. goodness of fit | 0.97 | 0.97 | 0.87 | 0.93 |
| AIC | 6902 | 5633 | 7014 | 4291 |

*$p < 0.05$, **$p < 0.01$, ***$p < 0.001$.

Therefore, as an exploratory extension to our pre-registered analysis, we considered a maximally simple model with only the path through cognitive performance included. These performance-only models fit extremely well for both relative ($\chi^2_{(1)} < 1$, $\chi^2_{\text{robust}(1)} < 1$, adjusted goodness of fit = 1.0, AIC = 2748) and absolute estimation error ($\chi^2_{(1)} < 1.14$, $p = 0.29$, $\chi^2_{\text{robust}(1)} = 1.23$, $p = 0.27$, adjusted goodness of fit = 0.97, AIC = 2858), with total path strengths of −0.57 (95% CI (−0.67, −0.47)) and −0.31 (95% CI (−0.43, −0.20)), respectively. (The total path was of course equal to the only component path, via cognitive performance.)

To compare these performance-only models with the reduced models in table 3, we used AIC, an index of the quality of statistical models that takes account of model complexity [29]. The AIC increment between models (ΔAIC) can be converted into a relative likelihood for the model with the lower AIC, using the formula (exp(ΔAIC/2)). It is sometimes taken as a rule of thumb to prefer one model over another if ΔAIC is 2 units or more (relative likelihood greater than or equal to 2.72). For relative and absolute estimation error respectively, ΔIC compared with the corresponding reduced model was 2885 (= 5633–2748) and 1433 (= 4291–2858). In both cases, the relative likelihood of the performance-only model over the reduced model would be astronomically large (practically infinite). Although this was not part of our pre-registered plan, the observed improvement in model quality was not marginal, but decisive. This gives strong empirical grounds to prefer an account of the DKE driven by differences in performance between skilled and unskilled people, with no role for metacognitive factors (at least as we have measured them).

# 4. Discussion

## 4.1 Main outcomes

This Registered Report has examined the dual-burden account of the DKE for a test of matrix reasoning, the kind of complex cognitive task for which the DKE was first demonstrated [2]. The dual-burden account states that the greater self-overestimation among less-skilled people arises because they perform poorly and lack the metacognitive insight to realize it. We tested the assumptions of the dual-burden account and examined the role of metacognitive factors in driving the DKE. Metacognitive sensitivity tracked performance closely, so less information was exploited by the metacognitive judgements of poor performers. Metacognitive efficiency was not associated with performance, so the quality of metacognitive processing was not lower among poor performers. Metacognitive bias was positively associated with performance, so poor performers were less confident—not more confident—than good performers. A path analysis showed an overwhelming dominance of performance score in driving the DKE, and the influence of metacognitive factors was weakly against the DKE, or negligible. The assumptions of the dual-burden account may be partially satisfied, but metacognitive factors do not meaningfully drive the DKE.

## 4.2. Cognition and metacognition

The dual-burden account proposes that the DKE arises for tasks in which the judgement of one's own level of performance depends upon the same resources as the cognitive response itself. If an ideal observer has the same information available for cognition and metacognition, their metacognitive sensitivity ($meta$-$d'$) to discriminate correct from incorrect responses should be the same as their cognitive sensitivity ($d'$) for the task [16]. We would then expect $meta$-$d'$ to track $d'$ closely, albeit with some noise in the measures, as observed (figure 4$a$). This observed relationship is consistent with the dual-burden account, but it is relatively uninteresting at a psychological level, because (in the absence of additional task feedback) the information exploited by metacognition will always be limited by the information available for cognition. If a person has insufficient information to know the correct answers, they will also have insufficient information to know which of their answers are correct: 'It would be strange (based on the ideal observer model) if worse performing subjects didn't make noisier ratings' [12, p. 7].

However, the original dual-burden account was stated not just in terms of shared information, but as an overlap between the cognitive and metacognitive processes themselves: 'In essence, we argue that the skills that engender competence in a particular domain are often the very same skills necessary to evaluate competence in that domain…' [2, p. 1121]. This stronger claim implies that the quality of metacognitive processing should improve with task skill, predicting a positive relationship between performance and metacognitive efficiency. We found no evidence for this predicted relationship (figure 4$b$). The data support the idea that cognitive and metacognitive responses use common information, but not that they have overlapping mechanisms.

So unskilled people are less able to distinguish their successes and failures in matrix reasoning. However, the dual-burden story then makes a less secure inference, by casting the metacognitive insensitivity of unskilled people as a lack of insight into their failings, which makes them overconfident. This is questionable, because metacognitive insensitivity does not imply a specific insensitivity to failure any more than it implies a specific insensitivity to success. To examine the idea that unskilled people are overconfident we should consider the levels of confidence actually expressed.

## 4.3. Competence and confidence

Metacognitive sensitivity means that a person's confidence tracks their likelihood of success, so that they are more confident when responding correctly than when responding incorrectly. A simple measure of mean confidence would be biased by the rate of correct responding, especially for people with high metacognitive sensitivity. To adjust for the rate of correct responding, the measure of metacognitive bias is calculated as the unweighted mean confidence across correct and incorrect responses (table 1). Metacognitive bias was positively related to skill and performance (table 2 and figure 4$c$), so unskilled performers were less and not more confident than their competent counterparts, even adjusting for performance.

This positive relationship between competence and confidence is further amplified if we look at unadjusted mean confidence. Figure 5$a$ shows that the poorest performers are much less confident than the highest performers on this metric. Nonetheless, a proponent of the DKE might still ask whether the poor performers are as unconfident as they should be. The top performers score nearly perfectly, so their mean confidence rating should ideally be close to the top of the scale, which it is. The poorest performers score barely above chance, so their mean confidence rating should ideally be close to the bottom of the scale, which it is not. Instead, they have a mean confidence rating of around 2.3, just below the middle of the scale. If the dotted diagonal line represents the ideal confidence rating, then the high performers look appropriately confident, and the low performers look inappropriately overconfident, a pattern reminiscent of the classic DKE.

We would argue strongly against this interpretation of the confidence data. As figure 5$b$ shows, the overall mean confidence of the poorest performers (2.3) is at the same level as the mean confidence of a top-performer responding incorrectly. Top performers have high metacognitive sensitivity, so their ratings for incorrect responses reflect their confidence when they know they are uncertain. The fact that this confidence level is closer to the middle of the scale than the bottom may just be quirk of how people tend to use the scale; people may intuitively regard the lowest rating as high confidence that the response is incorrect (a conceptual mirror to using the top of the scale for high confidence that the response is correct). Empirically, the mean confidence of the poorest performer is equivalent to that of a top-performer facing a problem that is too hard for them. So, not only are low performers less confident than high performers, but they are less confident by an appropriate amount.

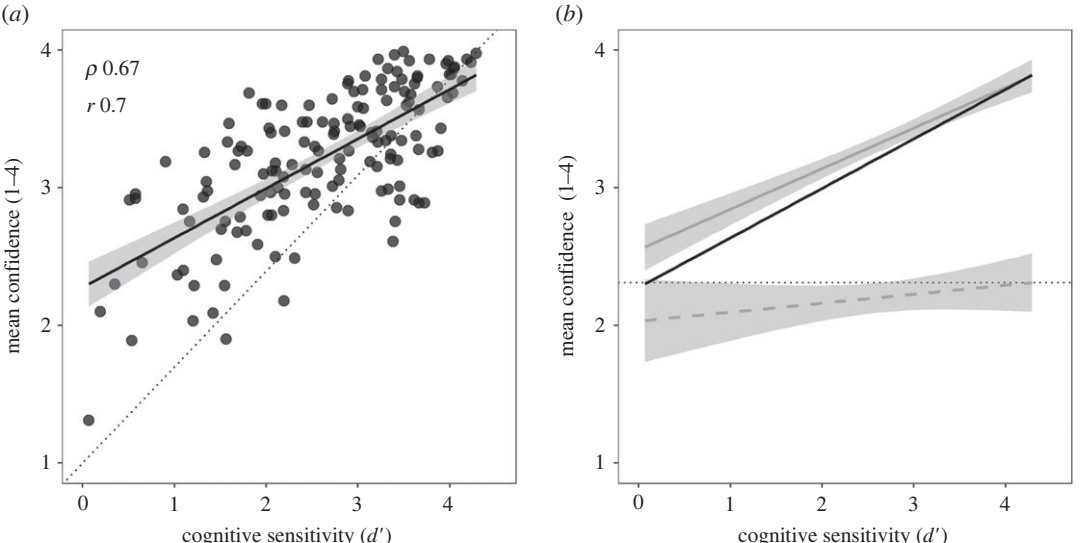

**Figure 5.** (a) Mean confidence as a function of cognitive sensitivity (a psychophysical measure of task performance). Unlike metacognitive bias (figure 4c), mean confidence is not adjusted for level of performance, so confidence is strongly related to performance. The dotted diagonal is the theoretically ideal confidence rating across the range of performance. Relative to this ideal, poor performers may appear to be overconfident. (b) The solid and dashed grey lines show mean confidence for correct and incorrect responses, respectively (as in figure 4d). The solid black line shows mean confidence (as in (a)), which is halfway between the grey lines for the poorest performers (who get around 50% correct) and converges to the solid grey line for the top performers (who get around 100% correct). The dotted horizontal line illustrates that the overall mean confidence of poor performers is equal to the mean confidence of top performers when they are incorrect. Relative to this benchmark, poor performers' overall level of confidence can be seen to be appropriate.

## 4.4. What drives the Dunning–Kruger effect?

The next question is whether metacognitive factors play an important role in creating the DKE. We replicated the DKE with negative correlations between task skill and self-estimation error, which were stronger for relative self-estimates ($\rho = -0.57$, $r = -0.53$) than for absolute self-estimates ($\rho = -0.43$, $r = -0.42$). We then used path analysis to quantify the respective roles of performance and metacognitive factors in mediating these relationships (cf. [11]). The DKE was overwhelmingly driven by performance scores, and the influence of metacognitive factors was weakly against the DKE, or negligible.

The lack of a driving role for metacognitive factors is not surprising, because it has long been argued that the DKE is mainly due to methodological artefacts (for a recent exchange, see [30–32]). As discussed in §1.1, one such artefact is regression to the mean due to double-dipping [4–7,11,33]. This double-dipping occurs when the same estimate of performance is used to index task skill and for the calculation of estimation errors, but it can be avoided by using an independent block of trials to estimate task skill, as we did. This will eliminate regression to the mean related to error in the measurement of performance,[10] but it will not neutralize other regressive biases, related to noise in the self-estimates.

People are generally rather poor at estimating their own performance, and their global self-estimates tend to be weakly related to objective performance: in the present study, relative self-estimates correlated with performance at $r = 0.43$, and absolute self-estimates correlated at $r = 0.47$. So self-estimates are uncertain and they tend to regress away from the extremes of the scale towards intermediate values. Given this noisy self-estimation, the main systematic influence on estimation error is the performance score itself, which is subtracted from the self-estimate to calculate estimation error. Performance is also positively related to task skill, so a negative relationship between task skill and estimation error (the DKE) is created. We previously called this dynamic a 'performance artefact' [11], because it is driven strongly by variations in the performance score, but it can also be understood as a form of regression to the mean, in which uncertain self-estimates regress towards intermediate values [4].

---

[10]The DKE pattern would have been stronger if we had correlated estimation error with performance in the same (test) block (as has often been done, following [2]). The correlations would then have been $\rho = -0.78$, $r = -0.63$ for relative estimation error, and $\rho = -0.49$, $r = -0.54$ for absolute estimation error. This confirms that regression to the mean caused by double-dipping acts to inflate the DKE (see §1.1)

As Burson and colleagues [4] emphasized in their 'noise-plus-bias' account of the DKE, the intermediate value to which self-estimates regress may be higher or lower than the true mean performance, creating asymmetrical patterns of regression. If the mean self-estimate is higher than mean performance, overestimation at the bottom end will be greater than underestimation at the top end, which is the prototypical form of the DKE. The prototypical DKE is quite reliably obtained with relative self-estimates, because—across a wide range of tasks—relative self-estimates tend to be optimistic. This optimism is sometimes known as the better-than-average effect: people routinely rate themselves on average above the 50th percentile, whereas the true mean is 50 by definition (see [34] for a recent review and meta-analysis). In the present study, the mean relative self-estimate was 58.3, so the better-than-average effect was present and the prototypical form of the DKE was obtained (figure 3c). The mean absolute self-estimate (62.4) was also above the middle of the scale, but in this case was lower than the mean performance (77.5), so the prototypical asymmetry was flipped, with underestimation at the top end exceeding overestimation at the bottom end (figure 3d). The *relatively* low mean absolute estimate may have been because participants failed to take full account of practice, which boosted scores in the test block (mean 77.5) relative to the baseline block (mean 55.8). Although participants were prompted to estimate *performance during the main phase only*, their absolute ratings may have been anchored by an impression of task difficulty from the baseline block.

## 4.5. Conclusion

Our study provides partial support for the assumptions of the dual-burden account, in that people who were less skilled at matrix reasoning also had lower metacognitive sensitivity, meaning that they were less able to discriminate their correct and incorrect responses. This suggests that cognitive and metacognitive processes have access to the same information. However, we found no evidence that the metacognitive processes of unskilled people were any less good than those of others, just that they had access to less good information. Nor were the poor performers overconfident; in fact, they were less confident than high performers, just as they should be. Crucially, the observed metacognitive differences do not drive the DKE. These data refute the dual-burden account, but they are compatible with a noise-plus-bias account, in which uncertainty makes self-estimates regressive with respect to performance, and general biases towards optimistic or pessimistic self-estimation make the regression patterns asymmetrical [4].

This conclusion should not be surprising, because global self-estimates are not valid measures of metacognition, and so are unlikely to support inferences about metacognition [12]. It is possible that some factors related to task skill could influence self-estimates and explain a minor proportion of the DKE [35], but the metacognitive variables that we have studied do not drive the DKE; if anything, they slightly counteract it. The present findings are very similar to our earlier findings for movement and memory tasks [11], suggesting that this conclusion is quite general. Indeed, the diverse range of tasks for which the DKE can be demonstrated is itself suggestive of a very general phenomenon like regression to the mean, rather than a specific hypothesis like the dual-burden account. We suspect that the DKE pattern might arise for literally any task, cognitive or otherwise, in which it is hard for people to know their score.

The present experiment confirms, for a matrix reasoning task, that unskilled people do lack insight, in the limited sense that they are less able to discriminate between correct and incorrect responses. But this does not mean that they are lacking metacognitive skill, or are overconfident, and the metacognitive differences that exist do not underlie the DKE. These findings clearly refute the dual-burden account of the DKE. This is important, because the dual-burden account has had wide cultural influence, being used to push harmful stereotypes of poor performers as blind to their own faults, arrogant and overconfident (*stupid people are too stupid to know they are stupid*). There may be some circumstances in which low competence is systematically accompanied by high confidence, but these would not be typical cases in which the task representation was weak or noisy, but special cases in which a person had a mistaken task representation so that they were systematically following the wrong rule to give incorrect answers with high confidence (e.g. [36]). Special cases aside, the DKE is an asymmetrical regression artefact that tells us nothing much about metacognition. Our understanding of metacognition will develop further and faster using more appropriate, theoretically grounded research methods [12].

**Ethics.** This study was conducted in accordance with the principles expressed in the Declaration of Helsinki, and informed consent was obtained from all participants. The study was approved by the PPLS Research Ethics Committee of the University of Edinburgh (approval numbers 422-1819/1; 422-1819/2).

Data accessibility. Compiled and source LabVIEW code for the experimental tasks, and full raw and processed data and analysis R code are archived at https://osf.io/u8kt4/. The original accepted Stage 1 Registered Report is archived at https://osf.io/u8kt4/. Materials, data and code available at https://osf.io/u8kt4/.

The data are provided in electronic supplementary material [37].

Authors' contributions. R.D.M.: conceptualization, data curation, formal analysis, methodology, resources, software, supervision, visualization, writing—original draft, writing—review and editing; A.B.M.: conceptualization, formal analysis, methodology, resources, supervision, writing—review and editing; Y.L.: data curation, investigation, project administration, writing—review and editing; S.D.S.: conceptualization, methodology, writing—review and editing.

All authors gave final approval for publication and agreed to be held accountable for the work performed therein.

Conflict of interest declaration. We declare we have no competing interests.

Funding. No funding external to the University of Edinburgh supported this research.

Acknowledgements. We are grateful to Karim Rivera Lares for assistance with pilot data collection.

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
