## [Peer Review File · Royal Society Open Science]

Review History

RSOS-191727.R0 (Original submission)

Review form: Reviewer 1 (Patrick R. Heck)

Do you have any ethical concerns with this paper?

No

Recommendation?

Major revision

Comments to the Author(s)

Thank you for inviting me to review “Skill and self-knowledge: do unskilled people really lack insight?” submitted as a proposal for in-principal acceptance as a Stage 1 Registered Report. I’m interested in measuring overconfidence, the better-than-average effect, and the Dunning-Kruger effect (DKE), and so I read this manuscript with great interest. Although I am familiar with Registered Reports (RR), I would like to disclose that this will be my first review for a manuscript

of this type; please forgive any comments in this review that may be overly speculative or inappropriate for the RR format.

Let me start with the manuscript's strengths. First, the question is important. Disentangling bias from error, and statistical artifacts from psychological phenomena, is a challenging and currently unresolved task in overconfidence and self-enhancement research. Some scholars have started moving toward psychophysical approaches to studying overconfidence and metacognition, and there is no question that the proposed manuscript can make a contribution with its objective task, repeated-measures design, and attempt to separately measure cognitive ability and metacognition. The theoretical framework is strong, and I believe the measure of metacognitive efficiency is both new and useful. Finally, the writing is concise, clear, and enjoyable.

Despite these strengths, I finished two read-throughs with several points of confusion and unanswered questions. My broad assessment is that the proposal, as written, sacrifices completeness for conciseness. The hypotheses may be valid, but at present, I am not certain that I fully understand them, their interrelations, or the measures used to test them. Although I do not see any fatal flaws in the design, and I believe the proposed dataset would advance our understanding of self-insight and the DKE, I think this manuscript would benefit from several points of expansion and clarification. I will detail these below. Following that, I will list additional, more minor concerns.

1) A more detailed explanation of the hypotheses, possibly to include an analysis of the pilot data the authors mention.

First, the cognitive and metacognitive sensitivity measures are explained simply as "computed using a toolbox" or package. Because these measures are critical to the hypotheses and overall research question, I think the manuscript would benefit from an explanation of how these measures are calculated (either conceptually, mathematically, or both). An audience of social psychologists, who are interested in this type of work, will likely not have an expert or intuitive understanding of some of the terms of signal detection theory.

Second, the hypotheses are not independent from one another (i.e., they all rely on cognitive sensitivity and some aspect of metacognition). They don't have to be independent, but the authors should explain more clearly what each hypothesis contributes or explains that the other hypotheses don't.

Third, it was quite difficult for me to keep in mind the differences between the authors' critical terms (sensitivity, bias, efficiency, etc). I think readers would find this section much clearer if the authors were to walk through the pilot data analysis, or possibly an analysis of simulated data (including a figure or table of the critical measures).

Finally, I'm not convinced that the test in Hypothesis 1 of a deviation from the identity line (in slope OR intercept) is really an important one. Would anyone really predict that participants' performance will perfectly predict their metacognitive assessment of this performance (or vice versa)? Furthermore, since accuracy is a binary measure (correct or not) and confidence is on a four-point scale, it seems even less plausible to me that the resulting measures of cognitive and metacognitive sensitivity could possibly both fit on the identity line in an empirical sample. I am not sure that null-hypothesis testing against this implausible null hypothesis is useful, and would suggest instead that the authors simply propose to estimate this relationship (and the size of any observed DKE) in their model.

2) Additional justifications for the thresholds and benchmarks chosen (even if those justifications are just admissions that a benchmark was chosen arbitrarily).

The authors state several benchmarks and thresholds for their predictions, but often refrain from explaining whether these choices were made based on data, theory, or arbitrary heuristics. Because nobody can predict exactly what these benchmarks should be, it seems perfectly acceptable to me to disclose when and where these choices were made without previous research or data to back them up.

3) An explanation of the population to be sampled.

I am assuming that the research population will be university students, but I did not see a description or disclosure of the target sample. Regardless of where the participants come from, it may also strengthen the proposal to discuss how and why the results may generalize outside of the population being studied. This also raises the question of whether participants' comparative estimates (performance percentile ranking estimate) will be made in relation to the average student, the average person of X nationality, or something else. This should be clarified.

4) A commitment to data openness and reproducibility (code; data; materials to be posted) (apologies if I missed this somewhere in the submission).

Given that pilot data have already been collected and analyzed, why are the data and code not uploaded with this submission? Do the authors plan to use the same analysis code from the pilot data to analyze the full sample? Preregistering and uploading this code would aid in reducing flexibility in analysis after all the data are collected. And if analyses change, or if the authors decide to conduct an exploratory analysis after the fact using new or modified code, they simply need to note this in the final version of the manuscript. Uploading the data, analysis code, and some of the results from the pilot data would also aid in my point #1 above.

Finally, I do feel strongly that a final version of this manuscript should include open data and reproducible code. Researchers in the future may wish to try competing models or test the robustness of the results to certain exclusion criteria. I encourage the authors to state that they would be willing to post code and data upon publication (or if not, why). Again, I apologize if the authors have already done this as part of the submission process and I missed it.

5) More justification for the path analysis.

The path analysis may be interesting, but the authors currently describe it as exploratory and leave it at that. What would it mean if certain paths were supported? Currently, I'm not sure this analysis is necessary: I think we need to see more discussion of competing predictions or possible outcomes to justify its inclusion.

6) More justification for the split between cognitive skill and cognitive performance.

After two read-throughs, I still don't understand why the authors propose to separate the trials into measures of skill and performance. Furthermore, they use this proposal as a way to refute the issue of regression to the mean as an explanation for inflate self-estimates (p. 12), following this claim up with a single large block of citations. If they are essentially the same puzzles, and they are shuffled throughout the paradigm, then how are the test trials free from the constraint of regression to the mean, simply by nature of being separate from the baseline trials? If the test trials are essentially just transformations or proxies for the baseline trials, then it isn't clear to me how these two measures (skill and performance) avoid regression effects. The authors clearly believe that this is the right approach, but the logic is not clear to me, and I think it may also confuse other potential readers. This is another area where the manuscript's conciseness comes at a cost of clarity and careful explanation of the underlying logic.

Minor points:

(Please pardon the self-citation) it may be useful to consult Heck & Krueger (2015, JEP:G) for a decision-theoretic analysis of error and bias in the claim to be above average (or not) after completing a single task.

I have two comments on the proposed samples size and power analysis. First, the authors used a corrected alpha level of .01, which seems appropriate, but I did not see an explicit statement that the authors will use this as the criterion for their statistical tests. Second, I suspect that a substantial number of participants will have heard of the DKE before (especially if they are psychology students) – perhaps more than the authors allow for in their power analysis. I appreciate the commitment to analyze the data before and after exclusion, and I also think it may be useful to examine the DKE in those participants who report having knowledge of it.

I think I may be missing something in Hypothesis 2: the authors predict a correlation between d' and $(\text{meta-}d'/d')$, but the latter term contains the former, and so they must be correlated. As d' increases, then the fractional term must decrease. Is the hypothesis that the observed correlation will differ from this analytical truth?

For outcome-neutral condition #2, the term “and/or” suggests flexibility in how to proceed if the resulting correlations do not both achieve .3. I share the authors’ prediction that these correlations will exceed .3 for absolute and relative EE, but the language should be more explicit about what would happen if one correlation was above .3 and another was below .3.

Thank you for your work; I look forward to the future of this manuscript.

Review form: Reviewer 2 (Steve Fleming)

Do you have any ethical concerns with this paper?

No

Recommendation?

Major revision

Comments to the Author(s)

This Stage 1 registered report sets out to address a foundational question in metacognition research – the extent to which the celebrated Dunning-Kruger effect (DKE) is a true metacognitive deficit, or not. This is a timely question that is well suited to psychophysical assays of metacognition that do not suffer from the ambiguities in interpretation of the one-shot measures in the classic DKE studies. The study proposal is state-of-the-art in this regard, and I think the authors are well placed to make a definitive contribution to this literature. I mainly have comments about the design and analysis approach that I hope will be helpful as the authors finalise their study plans.

1) My main concern is that the adaptation of the abstract reasoning task adapts it from 4-AFC to a yes/no task – i.e. asking the subject is the solution correct, or not. Moving away from 4-AFC is necessary to allow estimation of meta- d' , but I worry that the yes/no version introduces other problems, particularly that response-conditional asymmetries in meta- d' may interact with the hypotheses under test. In both perceptual and memory experiments, it is well-known that metacognitive sensitivity following “no” responses (i.e. responding that a stimulus was absent, or

the word was new/not on the list in recognition memory experiments) is substantially worse than metacognitive sensitivity following “yes” responses (Fleming & Dolan, 2010 *Consciousness & Cognition*; Kanai et al., 2010 *Consciousness & Cognition*; Higham et al., 2009 *JEP:LMC*; Meuwese et al. 2014 *AP&P*). Responding correct/incorrect is formally equivalent to responding yes/no (and the response is framed as Y vs. N), and therefore I worry it will lead to similar metacognitive asymmetries which are orthogonal to the main question of interest. To avoid this why not use a 2AFC task – i.e. the subject needs to respond which out of two options is the correct solution? If there is a strong reason why the yes/no version needs to be used, response-conditional estimation of meta- d' should be employed to allow for these asymmetries in the type 2 ROC fit. If this is not done, then the standard meta- d' model will be unable to accommodate response-conditional divergence between the two type 2 ROC curves and model fits will be poor (https://github.com/metacoglab/HMeta-d/blob/master/Matlab/exampleFit_rc.m for a simulation of this).

2) A second concern is conceptual. The dual-burden hypothesis is clearly stated in the introduction – that similar psychological resources contribute to both performance and metacognition, such that the worst performers are also the metacognitively least able. But the way this hypothesis relates to the constructs of metacognitive sensitivity (meta- d') and metacognitive efficiency (meta- d'/d') is less clear. On p. 16, it's stated that the dual-burden hypothesis predicts an identity between meta- d' and d' . This is fine, but also psychologically weak, as this relationship would also fall out of a first-order SDT model with no metacognitive “capacity” to speak of. Indeed, this is why the meta- d' framework is useful, because it allows correcting for this influence of performance on measures of metacognition. Instead, a psychologically stronger form of the dual burden hypothesis, and the one that seems most aligned to a metacognitive interpretation of the DKE, is that metacognitive efficiency, corrected for performance, scales with d' . This is discussed at the top of p. 17, but I found it odd that it was not considered a test of the general dual burden idea.

Minor points:

- The authors might want to consider using the metacognitive threshold measure proposed by Sherman et al. (<https://www.biorxiv.org/content/10.1101/361543v1>) rather than metacognitive bias, as this has more stable psychometric properties and is independent of variation in meta- d' .
- p. 13, the HMeta-d toolbox employs Bayesian parameter estimation using MCMC sampling, not maximum likelihood estimation.
- Will the HMeta-d toolbox be used in single-subject or group estimation mode? If the latter, note that single-subject parameter estimates will be shrunk to the group mean, which is less useful for subsequent correlation analyses. Instead, a powerful approach is to embed tests of correlations within a hierarchical regression (see https://github.com/metacoglab/HMeta-d/blob/master/Matlab/exampleFit_group_regression.m). But given that the current design has a decent number of trials per subject (180), sticking to single-subject Bayesian estimation may be sufficient and more straightforward.

Signed: Steve Fleming

Decision letter (RSOS-191727.R0)

11-Nov-2019

Dear Professor McIntosh,

The Editors assigned to your Stage 1 Registered Report ("Skill and self-knowledge: do unskilled people really lack insight?") have now received comments from reviewers. We would like you to revise your paper in accordance with the referee and editors suggestions which can be found below (not including confidential reports to the Editor). Please note this decision does not guarantee eventual acceptance.

Please submit a copy of your revised paper within three weeks (i.e. by the The author due date is unavailable) or let the journal office know if you would like an extension.

When submitting your revised manuscript, you must respond to the comments made by the referees and upload a file "Response to Referees" in "Section 2 - File Upload". Please use this to document how you have responded to the comments, and the adjustments you have made. In order to expedite the processing of the revised manuscript, please be as specific as possible in your response.

Kind regards,
Andrew Dunn
Senior Publishing Editor
Royal Society Open Science
openscience@royalsociety.org

on behalf of Professor Chris Chambers (Registered Reports Editor, Royal Society Open Science)
openscience@royalsociety.org

Associate Editor Comments to Author (Professor Chris Chambers):

Two expert reviewers have now appraised the manuscript. Both reviews are encouraging overall, while also raising a number of substantial issues that will need to be addressed in revision. To summarise the key points: Reviewer 1 questions the rationale of the hypotheses and notes a number of areas where greater clarity and methodological detail are required. The reviewer also points out the importance of open data/code -- this is already a policy requirement for RSOS but please do respond affirmatively on this point to reassure the reviewer. Reviewer 2 also questions the rationale for the hypotheses (and the dual-burden hypothesis especially) and questions where the formulation of this hypothesis is the most appropriate and severe test of the underlying theory. The reviewer also questions the suitability of the proposed abstract reasoning task. Overall the reviews, while critical, are sufficiently positive and constructive to place the manuscript firmly in the revisable range for a Stage 1 Registered Report. A Major Revision is therefore invited.

Comments to Author:

Reviewer: 1

Comments to the Author(s)

Thank you for inviting me to review “Skill and self-knowledge: do unskilled people really lack insight?” submitted as a proposal for in-principal acceptance as a Stage 1 Registered Report. I’m interested in measuring overconfidence, the better-than-average effect, and the Dunning-Kruger effect (DKE), and so I read this manuscript with great interest. Although I am familiar with Registered Reports (RR), I would like to disclose that this will be my first review for a manuscript of this type; please forgive any comments in this review that may be overly speculative or inappropriate for the RR format.

Let me start with the manuscript’s strengths. First, the question is important. Disentangling bias from error, and statistical artifacts from psychological phenomena, is a challenging and currently unresolved task in overconfidence and self-enhancement research. Some scholars have started moving toward psychophysical approaches to studying overconfidence and metacognition, and there is no question that the proposed manuscript can make a contribution with its objective task, repeated-measures design, and attempt to separately measure cognitive ability and metacognition. The theoretical framework is strong, and I believe the measure of metacognitive efficiency is both new and useful. Finally, the writing is concise, clear, and enjoyable.

Despite these strengths, I finished two read-throughs with several points of confusion and unanswered questions. My broad assessment is that the proposal, as written, sacrifices completeness for conciseness. The hypotheses may be valid, but at present, I am not certain that I fully understand them, their interrelations, or the measures used to test them. Although I do not see any fatal flaws in the design, and I believe the proposed dataset would advance our understanding of self-insight and the DKE, I think this manuscript would benefit from several points of expansion and clarification. I will detail these below. Following that, I will list additional, more minor concerns.

1) A more detailed explanation of the hypotheses, possibly to include an analysis of the pilot data the authors mention.

First, the cognitive and metacognitive sensitivity measures are explained simply as “computed using a toolbox” or package. Because these measures are critical to the hypotheses and overall research question, I think the manuscript would benefit from an explanation of how these measures are calculated (either conceptually, mathematically, or both). An audience of social psychologists, who are interested in this type of work, will likely not have an expert or intuitive understanding of some of the terms of signal detection theory.

Second, the hypotheses are not independent from one another (i.e., they all rely on cognitive sensitivity and some aspect of metacognition). They don’t have to be independent, but the authors should explain more clearly what each hypothesis contributes or explains that the other hypotheses don’t.

Third, it was quite difficult for me to keep in mind the differences between the authors’ critical terms (sensitivity, bias, efficiency, etc). I think readers would find this section much clearer if the authors were to walk through the pilot data analysis, or possibly an analysis of simulated data (including a figure or table of the critical measures).

Finally, I’m not convinced that the test in Hypothesis 1 of a deviation from the identity line (in slope OR intercept) is really an important one. Would anyone really predict that participants’ performance will perfectly predict their metacognitive assessment of this performance (or vice

versa)? Furthermore, since accuracy is a binary measure (correct or not) and confidence is on a four-point scale, it seems even less plausible to me that the resulting measures of cognitive and metacognitive sensitivity could possibly both fit on the identity line in an empirical sample. I am not sure that null-hypothesis testing against this implausible null hypothesis is useful, and would suggest instead that the authors simply propose to estimate this relationship (and the size of any observed DKE) in their model.

2) Additional justifications for the thresholds and benchmarks chosen (even if those justifications are just admissions that a benchmark was chosen arbitrarily).

The authors state several benchmarks and thresholds for their predictions, but often refrain from explaining whether these choices were made based on data, theory, or arbitrary heuristics. Because nobody can predict exactly what these benchmarks should be, it seems perfectly acceptable to me to disclose when and where these choices were made without previous research or data to back them up.

3) An explanation of the population to be sampled.

I am assuming that the research population will be university students, but I did not see a description or disclosure of the target sample. Regardless of where the participants come from, it may also strengthen the proposal to discuss how and why the results may generalize outside of the population being studied. This also raises the question of whether participants' comparative estimates (performance percentile ranking estimate) will be made in relation to the average student, the average person of X nationality, or something else. This should be clarified.

4) A commitment to data openness and reproducibility (code; data; materials to be posted) (apologies if I missed this somewhere in the submission).

Given that pilot data have already been collected and analyzed, why are the data and code not uploaded with this submission? Do the authors plan to use the same analysis code from the pilot data to analyze the full sample? Preregistering and uploading this code would aid in reducing flexibility in analysis after all the data are collected. And if analyses change, or if the authors decide to conduct an exploratory analysis after the fact using new or modified code, they simply need to note this in the final version of the manuscript. Uploading the data, analysis code, and some of the results from the pilot data would also aid in my point #1 above.

Finally, I do feel strongly that a final version of this manuscript should include open data and reproducible code. Researchers in the future may wish to try competing models or test the robustness of the results to certain exclusion criteria. I encourage the authors to state that they would be willing to post code and data upon publication (or if not, why). Again, I apologize if the authors have already done this as part of the submission process and I missed it.

5) More justification for the path analysis.

The path analysis may be interesting, but the authors currently describe it as exploratory and leave it at that. What would it mean if certain paths were supported? Currently, I'm not sure this analysis is necessary: I think we need to see more discussion of competing predictions or possible outcomes to justify its inclusion.

6) More justification for the split between cognitive skill and cognitive performance.

After two read-throughs, I still don't understand why the authors propose to separate the trials into measures of skill and performance. Furthermore, they use this proposal as a way to refute

the issue of regression to the mean as an explanation for inflated self-estimates (p. 12), following this claim up with a single large block of citations. If they are essentially the same puzzles, and they are shuffled throughout the paradigm, then how are the test trials free from the constraint of regression to the mean, simply by nature of being separate from the baseline trials? If the test trials are essentially just transformations or proxies for the baseline trials, then it isn't clear to me how these two measures (skill and performance) avoid regression effects. The authors clearly believe that this is the right approach, but the logic is not clear to me, and I think it may also confuse other potential readers. This is another area where the manuscript's conciseness comes at a cost of clarity and careful explanation of the underlying logic.

Minor points:

(Please pardon the self-citation) it may be useful to consult Heck & Krueger (2015, JEP:G) for a decision-theoretic analysis of error and bias in the claim to be above average (or not) after completing a single task.

I have two comments on the proposed sample size and power analysis. First, the authors used a corrected alpha level of .01, which seems appropriate, but I did not see an explicit statement that the authors will use this as the criterion for their statistical tests. Second, I suspect that a substantial number of participants will have heard of the DKE before (especially if they are psychology students) – perhaps more than the authors allow for in their power analysis. I appreciate the commitment to analyze the data before and after exclusion, and I also think it may be useful to examine the DKE in those participants who report having knowledge of it.

I think I may be missing something in Hypothesis 2: the authors predict a correlation between d' and $(\text{meta-}d'/d')$, but the latter term contains the former, and so they must be correlated. As d' increases, then the fractional term must decrease. Is the hypothesis that the observed correlation will differ from this analytical truth?

For outcome-neutral condition #2, the term “and/or” suggests flexibility in how to proceed if the resulting correlations do not both achieve .3. I share the authors' prediction that these correlations will exceed .3 for absolute and relative EE, but the language should be more explicit about what would happen if one correlation was above .3 and another was below .3.

Thank you for your work; I look forward to the future of this manuscript.

Reviewer: 2

Comments to the Author(s)

This Stage 1 registered report sets out to address a foundational question in metacognition research – the extent to which the celebrated Dunning-Kruger effect (DKE) is a true metacognitive deficit, or not. This is a timely question that is well suited to psychophysical assays of metacognition that do not suffer from the ambiguities in interpretation of the one-shot measures in the classic DKE studies. The study proposal is state-of-the-art in this regard, and I think the authors are well placed to make a definitive contribution to this literature. I mainly have comments about the design and analysis approach that I hope will be helpful as the authors finalise their study plans.

1) My main concern is that the adaptation of the abstract reasoning task adapts it from 4-AFC to a yes/no task – i.e. asking the subject if the solution is correct, or not. Moving away from 4-AFC is necessary to allow estimation of meta- d' , but I worry that the yes/no version introduces other problems, particularly that response-conditional asymmetries in meta- d' may interact with the hypotheses under test. In both perceptual and memory experiments, it is well-known that metacognitive sensitivity following “no” responses (i.e. responding that a stimulus was absent, or

the word was new/not on the list in recognition memory experiments) is substantially worse than metacognitive sensitivity following “yes” responses (Fleming & Dolan, 2010 *Consciousness & Cognition*; Kanai et al., 2010 *Consciousness & Cognition*; Higham et al., 2009 *JEP:LMC*; Meuwese et al. 2014 *AP&P*). Responding correct/incorrect is formally equivalent to responding yes/no (and the response is framed as Y vs. N), and therefore I worry it will lead to similar metacognitive asymmetries which are orthogonal to the main question of interest. To avoid this why not use a 2AFC task – i.e. the subject needs to respond which out of two options is the correct solution? If there is a strong reason why the yes/no version needs to be used, response-conditional estimation of meta- d' should be employed to allow for these asymmetries in the type 2 ROC fit. If this is not done, then the standard meta- d' model will be unable to accommodate response-conditional divergence between the two type 2 ROC curves and model fits will be poor (https://github.com/metacoglab/HMeta-d/blob/master/Matlab/exampleFit_rc.m for a simulation of this).

2) A second concern is conceptual. The dual-burden hypothesis is clearly stated in the introduction – that similar psychological resources contribute to both performance and metacognition, such that the worst performers are also the metacognitively least able. But the way this hypothesis relates to the constructs of metacognitive sensitivity (meta- d') and metacognitive efficiency (meta- d'/d') is less clear. On p. 16, it's stated that the dual-burden hypothesis predicts an identity between meta- d' and d' . This is fine, but also psychologically weak, as this relationship would also fall out of a first-order SDT model with no metacognitive “capacity” to speak of. Indeed, this is why the meta- d' framework is useful, because it allows correcting for this influence of performance on measures of metacognition. Instead, a psychologically stronger form of the dual burden hypothesis, and the one that seems most aligned to a metacognitive interpretation of the DKE, is that metacognitive efficiency, corrected for performance, scales with d' . This is discussed at the top of p. 17, but I found it odd that it was not considered a test of the general dual burden idea.

Minor points:

- The authors might want to consider using the metacognitive threshold measure proposed by Sherman et al. (<https://www.biorxiv.org/content/10.1101/361543v1>) rather than metacognitive bias, as this has more stable psychometric properties and is independent of variation in meta- d' .
- p. 13, the HMeta-d toolbox employs Bayesian parameter estimation using MCMC sampling, not maximum likelihood estimation.
- Will the HMeta-d toolbox be used in single-subject or group estimation mode? If the latter, note that single-subject parameter estimates will be shrunk to the group mean, which is less useful for subsequent correlation analyses. Instead, a powerful approach is to embed tests of correlations within a hierarchical regression (see https://github.com/metacoglab/HMeta-d/blob/master/Matlab/exampleFit_group_regression.m). But given that the current design has a decent number of trials per subject (180), sticking to single-subject Bayesian estimation may be sufficient and more straightforward.

Signed: Steve Fleming

Author's Response to Decision Letter for (RSOS-191727.R0)

See Appendix A.

RSOS-191727.R1 (Revision)

Review form: Reviewer 1 (Patrick R. Heck)

Do you have any ethical concerns with this paper?

No

Recommendation?

Accept with minor revision

Comments to the Author(s)

I found this revision to be extremely responsive to my comments and I no longer have any major concerns. Below, I am noting a few minor blemishes that should be fixed before acceptance. Otherwise, I am grateful to the authors for their thoughtful revisions and clear responses to my review. And I am very much looking forward to seeing the data.

Minor concerns:

In the introduction, two back-to-back paragraphs begin with a "however," clause, which is unsightly.

For the relative and absolute self-estimates, I noticed that one is on a 1-100 scale and the other is on a 0-100 scale. Unless the authors feel that it is theoretically or practically important to keep this difference, I would consider changing one of these so that they are both on the same scale.

The acronym "ART" (p. 26) does not seem to be introduced or defined -- could this be a carryover from the original submission?

I noticed a typo in the link in footnote 7: "souceforge"

Review form: Reviewer 2 (Steve Fleming)

Do you have any ethical concerns with this paper?

No

Recommendation?

Accept in principle

Comments to the Author(s)

The authors have comprehensively addressed my previous comments, and I look forward to seeing the results of their experiment.

Just one minor correction - as JAGS uses precision and not variance, the variance of the prior on meta-d' in the single-subject model is 2, not 0.5.

Decision letter (RSOS-191727.R1)

03-Apr-2020

Dear Professor McIntosh,

On behalf of the Editors, I am pleased to inform you that your Manuscript RSOS-191727.R1 entitled "Skill and self-knowledge: do unskilled people really lack insight?" has been accepted in principle for publication in Royal Society Open Science subject to minor revision in accordance with the referee and editor suggestions. Please find their comments at the end of this email.

The reviewers and handling editors have recommended publication, but also suggest some minor revisions to your manuscript. Therefore, I invite you to respond to the comments and revise your manuscript.

Please submit the revised version of your manuscript within 4 weeks (i.e. by the 11-Apr-2020). If you do not think you will be able to meet this date please let us know.

When submitting your revised manuscript, you will be able to respond to the comments made by the referees and you should upload a file "Response to Referees". You can use this to document any changes you make to the original manuscript. In order to expedite the processing of the revised manuscript, please be as specific as possible in your response to the referees.

Full author guidelines can be found here <https://royalsocietypublishing.org/rsos/registered-reports>.

Kind regards,
Lianne Parkhouse
Editorial Coordinator
Royal Society Open Science
openscience@royalsociety.org

on behalf of Chris Chambers (Subject Editor, Royal Society Open Science)
openscience@royalsociety.org

Associate Editor Comments to Author (Professor Chris Chambers):

Both reviewers have responded positively to the revision and note only a few minor issues to address. Provided the revised manuscript adequately addresses these remaining points, Stage 1 IPA will follow without requiring further in-depth review.

Reviewer: 1

Comments to the Author(s)

I found this revision to be extremely responsive to my comments and I no longer have any major concerns. Below, I am noting a few minor blemishes that should be fixed before acceptance.

Otherwise, I am grateful to the authors for their thoughtful revisions and clear responses to my review. And I am very much looking forward to seeing the data.

Minor concerns:

In the introduction, two back-to-back paragraphs begin with a "however," clause, which is unsightly.

For the relative and absolute self-estimates, I noticed that one is on a 1-100 scale and the other is on a 0-100 scale. Unless the authors feel that it is theoretically or practically important to keep this difference, I would consider changing one of these so that they are both on the same scale.

The acronym "ART" (p. 26) does not seem to be introduced or defined -- could this be a carryover from the original submission?

I noticed a typo in the link in footnote 7: "souceforge"

Reviewer comments to Author:

Reviewer: 2

Comments to the Author(s)

The authors have comprehensively addressed my previous comments, and I look forward to seeing the results of their experiment.

Just one minor correction - as JAGS uses precision and not variance, the variance of the prior on meta-d' in the single-subject model is 2, not 0.5.

Author's Response to Decision Letter for (RSOS-191727.R1)

See Appendix B.

Decision letter (RSOS-191727.R2)

08-Apr-2020

Dear Professor McIntosh

On behalf of the Editor, I am pleased to inform you that your Manuscript RSOS-191727.R2 entitled "Skill and self-knowledge: do unskilled people really lack insight?" has been accepted in principle for publication in Royal Society Open Science.

You may now progress to Stage 2 and complete the study as approved. Before commencing data collection we ask that you:

- 1) Update the journal office as to the anticipated completion date of your study.
- 2) Register your approved protocol on the Open Science Framework (<https://osf.io/>) or other recognised repository, either publicly or privately under embargo until submission of the Stage 2 manuscript. Please note that a time-stamped, independent registration of the protocol is

mandatory under journal policy, and manuscripts that do not conform to this requirement cannot be considered at Stage 2. The protocol should be registered unchanged from its current approved state, with the time-stamp preceding implementation of the approved study design.

Following completion of your study, we invite you to resubmit your paper for peer review as a Stage 2 Registered Report. Please note that your manuscript can still be rejected for publication at Stage 2 if the Editors consider any of the following conditions to be met:

- The results were unable to test the authors' proposed hypotheses by failing to meet the approved outcome-neutral criteria.
- The authors altered the Introduction, rationale, or hypotheses, as approved in the Stage 1 submission.
- The authors failed to adhere closely to the registered experimental procedures. Please note that any deviations from the approved experimental procedures must be communicated to the editor immediately for approval, and prior to the completion of data collection. Failure to do so can result in revocation of in-principle acceptance and rejection at Stage 2 (see complete guidelines for further information).
- Any post-hoc (unregistered) analyses were either unjustified, insufficiently caveated, or overly dominant in shaping the authors' conclusions.
- The authors' conclusions were not justified given the data obtained.

We encourage you to read the complete guidelines for authors concerning Stage 2 submissions at <https://royalsocietypublishing.org/rsos/registered-reports#ReviewerGuideRegRep>. Please especially note the requirements for data sharing, reporting the URL of the independently registered protocol, and that withdrawing your manuscript will result in publication of a Withdrawn Registration.

Please note that Royal Society Open Science will introduce article processing charges for all new submissions received from 1 January 2018. Registered Reports submitted and accepted after this date will ONLY be subject to a charge if they subsequently progress to and are accepted as Stage 2 Registered Reports. If your manuscript is submitted and accepted for publication after 1 January 2018 (i.e. as a full Stage 2 Registered Report), you will be asked to pay the article processing charge, unless you request a waiver and this is approved by Royal Society Publishing. You can find out more about the charges at <https://royalsocietypublishing.org/rsos/charges>. Should you have any queries, please contact openscience@royalsociety.org.

Once again, thank you for submitting your manuscript to Royal Society Open Science and we look forward to receiving your Stage 2 submission. If you have any questions at all, please do not hesitate to get in touch. We look forward to hearing from you shortly with the anticipated submission date for your stage two manuscript.

on behalf of Professor Chris Chambers (Registered Reports Editor, Royal Society Open Science)
openscience@royalsociety.org

Author's Response to Decision Letter for (RSOS-191727.R2)

See Appendix C.

RSOS-191727.R3

Review form: Reviewer 1 (Patrick R. Heck)

Is the manuscript scientifically sound in its present form?

Yes

Are the interpretations and conclusions justified by the results?

Yes

Is the language acceptable?

Yes

Do you have any ethical concerns with this paper?

No

Have you any concerns about statistical analyses in this paper?

No

Recommendation?

Accept as is

Comments to the Author(s)

This is a comprehensive, well-written Stage 2 report. I do not have any major concerns with the treatment, analysis, and reporting of the data, and I thank the authors for considering my comments in Stage 1.

I was able to access the underlying data, Readme, and reproducibility files in the online archive. I reviewed (but did not run) the R files, but I could not review the MATLAB files due to a lack of access to the program. For those files I was able to review, I am satisfied with the open data and reproducibility in this report.

-Whether the data are able to test the authors' proposed hypotheses by passing the approved outcome-neutral criteria (such as absence of floor and ceiling effects or success of positive controls)

Yes, the data pass the outcome-neutral criteria specified in Stage 1.

-Whether the Introduction, rationale and stated hypotheses are the same as the approved Stage 1 submission

Yes.

-Whether the authors adhered precisely to the registered experimental procedures

Yes.

-Where applicable, whether any unregistered exploratory statistical analyses are justified, methodologically sound, and informative

Yes, unregistered analyses were clearly marked.

-Whether the authors' conclusions are justified given the data

I believe they are. The authors appropriately contextualize the measure they use as limited in some ways. I agree that it is appropriate but limited, and I would like to see future research that reports extensions of these hypotheses and analyses into more complex, real-world domains (i.e., social decision tasks; interpersonal interaction, etc).

Review form: Reviewer 2 (Steve Fleming)

Is the manuscript scientifically sound in its present form?

Yes

Are the interpretations and conclusions justified by the results?

Yes

Is the language acceptable?

Yes

Do you have any ethical concerns with this paper?

No

Have you any concerns about statistical analyses in this paper?

No

Recommendation?

Accept with minor revision

Comments to the Author(s)

It's a pleasure to be able to review again this exemplary registered report on the Dunning-Kruger Effect (DKE). I was excited to see the results, and after re-reading the entire manuscript I continue to think that the paper will make a definitive contribution on the relationship between metacognition and the DKE.

The pre-registered sample size was achieved with minimal exclusions, and with only minor deviations from the pre-registered protocol due to Covid which were pre-approved before data collection. Notably, the outcome-neutral tests of the classic DKE signatures of absolute and relative estimation error were passed with effect sizes comparable to the published literature. This makes the subsequent investigation of potential links to task-based metrics of metacognition straightforward and compelling.

I have one comment on the description of the methods for fitting meta- d' , and two other minor comments on presentation:

1) The description of metacognitive sensitivity estimation on p. 18 is generally clear but there may be some misunderstanding of how d' is estimated in the toolbox. It's said that "cognitive sensitivity (d') was estimated via Bayesian inference", but this is not the case when

using the toolbox default settings (in the `mcmc_params` function, and in the defaults of `fit_meta_d_mcmc`, `mcmc_params.estimate_dprime = 0` is the default setting). Instead, a point estimate of d' is calculated using the standard formula applied to hit and false alarm rates, and used as the prior for Bayesian estimation of meta- d (it's slightly confusing as the generative model always includes the priors on type 1 d' estimation in the text file, but when `estimate_dprime = 0`, d' is treated as an observed variable that is passed into the model, and so becomes fixed and not estimated).

It's possible to do joint Bayesian estimation of both d' and meta- d' by setting `mcmc_params.estimate_dprime = 1`, but we decided to make this a user option rather than a default as it can lead to a lack of convergence especially with low d' values or small trial numbers. It's also closer to the two-step estimation implemented in the original Maniscalco & Lau code.

I checked the analysis code uploaded to OSF and as far as I can tell the defaults were used (in `WrapperSSKMetacog.m`). If I am right about this, I think this section needs rewriting along the following lines:

"Cognitive sensitivity (d') was estimated from first-order response data using the following formula:

$$d' = z(H) - z(FA)$$

where $z()$ is the inverse of the standard cumulative normal distribution function, and H and FA are the type 1 hit and false alarm rates for each subject.

Metacognitive sensitivity (meta- d') was estimated within the HMeta- d toolbox, via Bayesian single-subject estimation (Fleming, 2017). Bayesian estimation of meta- d' has several advantages over , including, but not limited to, lack of need for edge correction, automatic precision for estimates relative to amount of data available, and quantification of uncertainty around estimates (see Lee, 2008). As described... etc"

- 2) The terms metacognitive sensitivity, efficiency and bias are used in the abstract without further explanation, but these are likely to be unfamiliar to a broader readership interested in the DKE. Perhaps a brief definition (eg in parentheses) could be added to aid dissemination?
- 3) P. 17, the acronym "EE" for estimation error is not defined and seems not to be used further.
- 4) P. 29-30, I found it hard to keep track of the sign of effects in the description of the path model. It might be useful to remind the reader that cognitive performance "negatively" drives the DKE, while "poorer" metacognition (in total) paradoxically tends to counteract it.

Signed: Steve Fleming

Decision letter (RSOS-191727.R3)

Dear Professor McIntosh:

On behalf of the Editor, I am pleased to inform you that your Stage 2 Registered Report RSOS-191727.R3 entitled "Skill and self-knowledge: empirical refutation of the dual-burden account of the Dunning-Kruger effect." has been deemed suitable for publication in Royal Society Open

Science subject to minor revision in accordance with the referee suggestions. Please find the referees' comments at the end of this email.

The reviewers and Subject Editor have recommended publication, but also suggest some minor revisions to your manuscript. We invite you to respond to the comments and revise your manuscript. Below the referees' and Editors' comments (where applicable) we provide additional requirements. Final acceptance of your manuscript is dependent on these requirements being met. We provide guidance below to help you prepare your revision.

Please submit your revised manuscript and required files (see below) no later than 7 days from today's (ie 31st October) date. Note: the ScholarOne system will 'lock' if submission of the revision is attempted 7 or more days after the deadline. If you do not think you will be able to meet this deadline please contact the editorial office immediately.

on behalf of Professor Chris Chambers (Associate Editor)
(Registered Reports Editor, Royal Society Open Science)
openscience@royalsociety.org

Associate Editor Comments to Author (Professor Chris Chambers):

Associate Editor: 1

Comments to the Author:

I have now received two helpful Stage 2 reviews. As you will see, both are very positive and I the manuscript should be suitable for full acceptance after a round a minor revision. Reviewer 2 notes several points to improve clarity, and one point concerning the analysis that requires careful attention.

Comments to Author:

Reviewer: 1

Comments to the Author(s)

This is a comprehensive, well-written Stage 2 report. I do not have any major concerns with the treatment, analysis, and reporting of the data, and I thank the authors for considering my comments in Stage 1.

I was able to access the underlying data, Readme, and reproducibility files in the online archive. I reviewed (but did not run) the R files, but I could not review the MATLAB files due to a lack of

access to the program. For those files I was able to review, I am satisfied with the open data and reproducibility in this report.

-Whether the data are able to test the authors' proposed hypotheses by passing the approved outcome-neutral criteria (such as absence of floor and ceiling effects or success of positive controls)

Yes, the data pass the outcome-neutral criteria specified in Stage 1.

-Whether the Introduction, rationale and stated hypotheses are the same as the approved Stage 1 submission

Yes.

-Whether the authors adhered precisely to the registered experimental procedures

Yes.

-Where applicable, whether any unregistered exploratory statistical analyses are justified, methodologically sound, and informative

Yes, unregistered analyses were clearly marked.

-Whether the authors' conclusions are justified given the data

I believe they are. The authors appropriately contextualize the measure they use as limited in some ways. I agree that it is appropriate but limited, and I would like to see future research that reports extensions of these hypotheses and analyses into more complex, real-world domains (i.e., social decision tasks; interpersonal interaction, etc).

Reviewer: 2

Comments to the Author(s)

It's a pleasure to be able to review again this exemplary registered report on the Dunning-Kruger Effect (DKE). I was excited to see the results, and after re-reading the entire manuscript I continue to think that the paper will make a definitive contribution on the relationship between metacognition and the DKE.

The pre-registered sample size was achieved with minimal exclusions, and with only minor deviations from the pre-registered protocol due to Covid which were pre-approved before data collection. Notably, the outcome-neutral tests of the classic DKE signatures of absolute and relative estimation error were passed with effect sizes comparable to the published literature. This makes the subsequent investigation of potential links to task-based metrics of metacognition straightforward and compelling.

I have one comment on the description of the methods for fitting meta- d' , and two other minor comments on presentation:

1) The description of metacognitive sensitivity estimation on p. 18 is generally clear but there may be some misunderstanding of how d' is estimated in the toolbox. It's said that "cognitive sensitivity (d') was estimated via Bayesian inference", but this is not the case when using the toolbox default settings (in the `mcmc_params` function, and in the defaults of `fit_meta_d_mcmc`, `mcmc_params.estimate_dprime = 0` is the default setting). Instead, a point estimate of d' is calculated using the standard formula applied to hit and false alarm rates, and used as the prior for Bayesian estimation of meta- d (it's slightly confusing as the generative model always includes

the priors on type 1 d' estimation in the text file, but when `estimate_dprime = 0`, d' is treated as an observed variable that is passed into the model, and so becomes fixed and not estimated).

It's possible to do joint Bayesian estimation of both d' and meta- d' by setting `mcmc_params.estimate_dprime = 1`, but we decided to make this a user option rather than a default as it can lead to a lack of convergence especially with low d' values or small trial numbers. It's also closer to the two-step estimation implemented in the original Maniscalco & Lau code.

I checked the analysis code uploaded to OSF and as far as I can tell the defaults were used (in `WrapperSSKMetacog.m`). If I am right about this, I think this section needs rewriting along the following lines:

"Cognitive sensitivity (d') was estimated from first-order response data using the following formula:

$$d' = z(H) - z(FA)$$

where $z()$ is the inverse of the standard cumulative normal distribution function, and H and FA are the type 1 hit and false alarm rates for each subject.

Metacognitive sensitivity (meta- d') was estimated within the HMeta-d toolbox, via Bayesian single-subject estimation (Fleming, 2017). Bayesian estimation of meta- d' has several advantages over , including, but not limited to, lack of need for edge correction, automatic precision for estimates relative to amount of data available, and quantification of uncertainty around estimates (see Lee, 2008). As described... etc"

2) The terms metacognitive sensitivity, efficiency and bias are used in the abstract without further explanation, but these are likely to be unfamiliar to a broader readership interested in the DKE. Perhaps a brief definition (eg in parentheses) could be added to aid dissemination?

3) P. 17, the acronym "EE" for estimation error is not defined and seems not to be used further.

4) P. 29-30, I found it hard to keep track of the sign of effects in the description of the path model. It might be useful to remind the reader that cognitive performance "negatively" drives the DKE, while "poorer" metacognition (in total) paradoxically tends to counteract it.

Signed: Steve Fleming

===PREPARING YOUR MANUSCRIPT===

one version should clearly identify all the changes that have been made (for instance, in coloured highlight, in bold text, or tracked changes);

===PREPARING YOUR REVISION IN SCHOLARONE===

- Ensure that your data access statement meets the requirements at <https://royalsociety.org/journals/authors/author-guidelines/#data>. You should ensure that you cite the dataset in your reference list. If you have deposited data etc in the Dryad repository, please only include the 'For publication' link at this stage. You should remove the 'For review' link.
- If you are requesting an article processing charge waiver, you must select the relevant waiver option (if requesting a discretionary waiver, the form should have been uploaded, see 'File upload' above).
- If you have uploaded any electronic supplementary (ESM) files, please ensure you follow the guidance at <https://royalsociety.org/journals/authors/author-guidelines/#supplementary-material> to include a suitable title and informative caption. An example of appropriate titling and captioning may be found at https://figshare.com/articles/Table_S2_from_Is_there_a_trade-off_between_peak_performance_and_performance_breadth_across_temperatures_for_aerobic_scope_in_teleost_fishes_/3843624.

Author's Response to Decision Letter for (RSOS-191727.R3)

See Appendix C.

Decision letter (RSOS-191727.R4)

Dear Professor McIntosh:

I am pleased to inform you that your manuscript entitled "Skill and self-knowledge: empirical refutation of the dual-burden account of the Dunning-Kruger effect." is now accepted for publication in Royal Society Open Science.

Please remember to make any data sets or code libraries 'live' prior to publication, and update any links as needed when you receive a proof to check - for instance, from a private 'for review' URL to a publicly accessible 'for publication' URL. It is also good practice to add data sets, code and other digital materials to your reference list.

Royal Society Open Science is a fully open access journal. A payment may be due before your article is published. Our partner Copyright Clearance Center's RightsLink for Scientific Communications will contact the corresponding author about your open access options from the email domain @copyright.com (if you have any queries regarding fees, please see <https://royalsocietypublishing.org/rsos/charges> or contact authorfees@royalsociety.org).

on behalf of Professor Chris Chambers (Subject Editor).

Follow Royal Society Publishing on Twitter: @RSocPublishing
Follow Royal Society Publishing on Facebook:
<https://www.facebook.com/RoyalSocietyPublishing/>
Read Royal Society Publishing's blog:
<https://royalsociety.org/blog/blogsearchpage/?category=Publishing>

Appendix A

Responses to reviewers

Preliminary note: The Stage 1 manuscript has been revised (and improved) substantially in response to reviewer comments. Also, in the time since the original submission, the Abstract Reasoning Task on which our main task is based (then referenced as a preprint: Fuhrmann, Chierchia, Knoll, Sakhardande, & Blakemore, 2018), has been published (in Royal Society Open Science), under a revised name, the matrix reasoning item bank (Chierchia et al., 2019). We have therefore updated our references and terminology accordingly.

Reviewer: 1

Comments to the Author(s)

Thank you for inviting me to review “Skill and self-knowledge: do unskilled people really lack insight?” submitted as a proposal for in-principal acceptance as a Stage 1 Registered Report. I’m interested in measuring overconfidence, the better-than-average effect, and the Dunning-Kruger effect (DKE), and so I read this manuscript with great interest. Although I am familiar with Registered Reports (RR), I would like to disclose that this will be my first review for a manuscript of this type; please forgive any comments in this review that may be overly speculative or inappropriate for the RR format.

Let me start with the manuscript’s strengths. First, the question is important. Disentangling bias from error, and statistical artifacts from psychological phenomena, is a challenging and currently unresolved task in overconfidence and self-enhancement research. Some scholars have started moving toward psychophysical approaches to studying overconfidence and metacognition, and there is no question that the proposed manuscript can make a contribution with its objective task, repeated-measures design, and attempt to separately measure cognitive ability and metacognition. The theoretical framework is strong, and I believe the measure of metacognitive efficiency is both new and useful. Finally, the writing is concise, clear, and enjoyable.

Despite these strengths, I finished two read-throughs with several points of confusion and unanswered questions. My broad assessment is that the proposal, as written, sacrifices completeness for conciseness. The hypotheses may be valid, but at present, I am not certain that I fully understand them, their interrelations, or the measures used to test them. Although I do not see any fatal flaws in the design, and I believe the proposed dataset would advance our understanding of self-insight and the DKE, I think this manuscript would benefit from several points of expansion and clarification. I will detail these below. Following that, I will list additional, more minor concerns.

1) A more detailed explanation of the hypotheses, possibly to include an analysis of the pilot data the authors mention.

(a) First, the cognitive and metacognitive sensitivity measures are explained simply as “computed using a toolbox” or package. Because these measures are critical to the hypotheses and overall research question, I think the manuscript would benefit from an explanation of how these measures are calculated (either conceptually, mathematically, or both). An audience of social psychologists, who are interested in this type of work, will likely not have an expert or intuitive understanding of some of the terms of signal detection theory.

We have now rewritten and expanded Introduction Section 1.2 to provide a fuller conceptual explanation of these cognitive and metacognitive measures, which will hopefully be accessible to those without prior knowledge of signal detection theory. We have also rewritten and expanded Methods Section 2.4.2 to provide a more precise mathematical description for those that want it. We appreciate the encouragement to expand on our previously rather terse descriptions, and we hope that these two approaches satisfy the need for both accessibility and technical detail.

(b) Second, the hypotheses are not independent from one another (i.e., they all rely on cognitive sensitivity and some aspect of metacognition). They don't have to be independent, but the authors should explain more clearly what each hypothesis contributes or explains that the other hypotheses don't.

It is possible for metacognitive sensitivity, efficiency and bias to vary independently of one another. However, as the reviewer correctly points out, the *relationships* we will examine are not mutually independent, because they all share one variable in common (cognitive sensitivity); and one variable (metacognitive efficiency) is a ratio measure between two others in the set (cognitive sensitivity, metacognitive sensitivity). Our aim is to estimate the strength of relationships between cognitive and metacognitive measures of theoretical interest. A lack of complete independence is not a problem for this aim, but it may seem to be so if the correlation analyses are conceptualised as a series of hypothesis tests.

In revising our analysis approach, we have therefore taken heed of the reviewer's encouragement (point 1d, below) to move away from hypothesis testing, and towards an estimation approach. Methods Section 2.6 has been revised to motivate this approach, and Methods Section 2.7.3 has been completely rewritten to explain our estimation approach, and to contextualise the relevance of each relationship with respect to the dual-burden account, picking up on the introductory material in Section 1.3.

Note that, in thinking harder about these issues, we realised that the analysis of metacognitive bias would support the characterisation of poor performers as overconfident only if a negative correlation was observed (previously, we suggested that any non-positive relationship would support the idea). We have updated our predictions for metacognitive bias in the last paragraph of Section 1.3, and the penultimate paragraph of Section 2.7.3.

(c) Third, it was quite difficult for me to keep in mind the differences between the authors' critical terms (sensitivity, bias, efficiency, etc). I think readers would find this section much clearer if the authors were to walk through the pilot data analysis, or possibly an analysis of simulated data (including a figure or table of the critical measures).

We now provide a write-up of the pilot study in Supplementary material (S1), explaining how it informed the final design, and the ways in which the final experiment differs from the pilot. This includes a link to open data and analysis code, so that anyone who wishes to examine the analysis steps in detail can do so.

The fine detail of the extraction of the metacognitive variables, performed within the Hmeta-d toolbox, are not described, because there would be a serious danger of turning the present paper into a technical methods paper. However, we do provide references to competent

technical papers that describe this analysis fully (e.g. Fleming & Daw, 2017; Maniscalco & Lau, 2012, 2017), and we quote the link to the open code for Fleming's Hmeta-d toolbox.

As noted in response to the reviewer's point 1a, we have made changes to the Introduction and to the Methods to provide better conceptual and technical descriptions of our dependent measures. At the reviewer's suggestion, we have now also added a summary table of dependent measures for this study (Table 1, referenced from Methods Sections 1.2. and 2.4.2), to help the reader keep the distinctions between critical terms clear.

(d) Finally, I'm not convinced that the test in Hypothesis 1 of a deviation from the identity line (in slope OR intercept) is really an important one. Would anyone really predict that participants' performance will perfectly predict their metacognitive assessment of this performance (or vice versa)? Furthermore, since accuracy is a binary measure (correct or not) and confidence is on a four-point scale, it seems even less plausible to me that the resulting measures of cognitive and metacognitive sensitivity could possibly both fit on the identity line in an empirical sample. I am not sure that null-hypothesis testing against this implausible null hypothesis is useful, and would suggest instead that the authors simply propose to estimate this relationship (and the size of any observed DKE) in their model.

Although meta-d' is derived from an analysis of confidence ratings made on a four-point scale, it is in d' units, directly comparable to first-order d', so it is not inherently implausible that an identity relationship could exist between them. And a literal reading of the dual-burden model, as originally articulated by Kruger & Dunning, and subsequently repeated across several academic papers and popular articles, would in fact predict an identity relationship, because the assertion is that metacognitive insight depends on *exactly the same* processes/resources as cognitive performance. Here are two typical examples:

"In essence, we argue that the skills that engender competence in a particular domain are often the very same skills necessary to evaluate competence in that domain" (Kruger & Dunning 1999, p1121).

"This double-curse arises because, in many life domains, the act of evaluating the correctness of one's (or anyone else's) response draws upon the exact same expertise that is necessary in choosing the correct response in the first place. That is, in the parlance of psychological research, the skills needed to execute the meta-cognitive task of judging the accuracy of a response are precisely the same as those necessary for the cognitive task of producing an accurate response." (Dunning, 2011, p261)

In terms of Fleming & Daw's model of metacognition, this would be equivalent to a first-order model in which metacognitive responses are based on the same information as cognitive responses (see Introduction, section 1.3). Therefore, in answer to the question of whether anyone would really predict this identity relationship, we would say that Dunning (and Kruger) would predict it, or at least that it is a logical consequence of their assertion of identity between cognitive and metacognitive processes.

At the same time, we agree with the reviewer that the prediction may nonetheless not be an important one, because it does seem implausible that metacognition would access the information on which cognition is based without any decay or transformation. A strong relationship between cognitive and metacognitive sensitivity could probably be taken to

support the assumptions of the dual-burden account, even if the line departed from a strict line of identity. We therefore accept that null-hypothesis testing is not useful here, and we very much appreciate the encouragement to pursue an estimation approach, which we agree is more appropriate. This approach is now introduced and defended in methods Section 2.6, and fleshed out in Section 2.7.3.

2) Additional justifications for the thresholds and benchmarks chosen (even if those justifications are just admissions that a benchmark was chosen arbitrarily).

The authors state several benchmarks and thresholds for their predictions, but often refrain from explaining whether these choices were made based on data, theory, or arbitrary heuristics. Because nobody can predict exactly what these benchmarks should be, it seems perfectly acceptable to me to disclose when and where these choices were made without previous research or data to back them up.

In light of this reviewer's previous comment, we have revised our statistical approach, focusing more upon parameter estimation, and less upon significance testing. We have also documented a more complete analysis of our pilot data (Supplementary material S1), in order to help inform the minimum effect sizes targeted by our critical tests of the DKE, and the expected effect sizes for estimation steps. The most relevant changes are to the Power and Sample Size section (Section 2.6), which now appears immediately before the statement of the statistical plan (so that our rationale is clarified before the statistical plan is described), as opposed to afterwards (as in the previous draft).

3) An explanation of the population to be sampled.

I am assuming that the research population will be university students, but I did not see a description or disclosure of the target sample. Regardless of where the participants come from, it may also strengthen the proposal to discuss how and why the results may generalize outside of the population being studied. This also raises the question of whether participants' comparative estimates (performance percentile ranking estimate) will be made in relation to the average student, the average person of X nationality, or something else. This should be clarified.

We had originally intended not to restrict our participant population, but our recruitment opportunities do mean that there will be a strong bias towards testing members of the student population of the University of Edinburgh. We have decided to make this explicit, and to restrict testing to this student population. We now state this in Methods Section 2.1. In accordance with this change, we have also now rephrased the prompt for the relative global estimate, to specify the appropriate reference population: "*Out of 100 Edinburgh University students performing this task, how do you think you would rank, if 1 is poorest and 100 is best?*", as specified in Section 2.3.

We think that restricting our target population in this way is reasonable given that the DKE was originally established, and has been studied predominantly within University samples (the original report was based on four studies of Cornell students; Kruger & Dunning, 1999).

The generalisability of the results obtained from such samples is an important issue. For instance, it has been suggested that the ability distribution expected in a student demographic

may help to explain the DKE pattern itself (Krajc, Ortmann, Krajč, & Ortmann, 2008). These are issues that we intend to address in the Discussion section of the Stage 2 report.

4) A commitment to data openness and reproducibility (code; data; materials to be posted) (apologies if I missed this somewhere in the submission).

Given that pilot data have already been collected and analyzed, why are the data and code not uploaded with this submission? Do the authors plan to use the same analysis code from the pilot data to analyze the full sample? Preregistering and uploading this code would aid in reducing flexibility in analysis after all the data are collected. And if analyses change, or if the authors decide to conduct an exploratory analysis after the fact using new or modified code, they simply need to note this in the final version of the manuscript. Uploading the data, analysis code, and some of the results from the pilot data would also aid in my point #1 above.

Finally, I do feel strongly that a final version of this manuscript should include open data and reproducible code. Researchers in the future may wish to try competing models or test the robustness of the results to certain exclusion criteria. I encourage the authors to state that they would be willing to post code and data upon publication (or if not, why). Again, I apologize if the authors have already done this as part of the submission process and I missed it.

We commit to posting open data, code and materials, for this study. This is actually a condition of publication for Registered Reports at *Royal Society Open Science*, and we made a declaration to this effect at the point of submission, although this may not have been made visible to the reviewer.

Open code and data are now standard practice in our lab (see <https://osf.io/ccgsz/>), e.g. our prior study on the DKE (doi/10.1037/xge0000579) was published with open data and code (<https://osf.io/8wjck/>) and a detailed preregistration document (<https://osf.io/ccgsz/>), and the pre-review version is archived as a pre-print (<https://psyarxiv.com/czms3/>).

For the present submission, we now include a write-up of the pilot study in **Supplementary material (S1)**, with a link to open data and code (<https://osf.io/u8kt4/>). The pilot study does not include the path analysis planned for the main study, but this follows the methods of our earlier paper, for which open code is available at the above links.

5) More justification for the path analysis.

The path analysis may be interesting, but the authors currently describe it as exploratory and leave it at that. What would it mean if certain paths were supported? Currently, I'm not sure this analysis is necessary: I think we need to see more discussion of competing predictions or possible outcomes to justify its inclusion.

The reviewer's comment makes it clear that we were not explicit enough about the role of the path analysis. The path analysis plays a critical role in directly modelling the dual-burden account, and in quantifying the relative influence of cognitive and metacognitive factors in driving the DKE. We have made several adjustments to make the purpose and importance of this analysis more evident.

In the **penultimate paragraph of Section 1.1**, we now expressly highlight the critical role of the equivalent path analysis in informing the conclusions of our previous study. This should prime the reader to understand the role of the path analysis in the present study. We have also rewritten **Section 2.7.4**, to more explicitly state the purpose of the path analysis, as distinct from the preceding analyses, referring back to the previous study (as reviewed in Section 1.1.). We also state how our conclusions will be informed by the possible outcomes.

Note that we previously suggested that we would use AIC and BIC for model selection. To remove any ambiguity about how we would act if these measures were not in agreement, we now propose to rely exclusively on AIC.

6) More justification for the split between cognitive skill and cognitive performance.

After two read-throughs, I still don't understand why the authors propose to separate the trials into measures of skill and performance. Furthermore, they use this proposal as a way to refute the issue of regression to the mean as an explanation for inflated self-estimates (p. 12), following this claim up with a single large block of citations. If they are essentially the same puzzles, and they are shuffled throughout the paradigm, then how are the test trials free from the constraint of regression to the mean, simply by nature of being separate from the baseline trials? If the test trials are essentially just transformations or proxies for the baseline trials, then it isn't clear to me how these two measures (skill and performance) avoid regression effects. The authors clearly believe that this is the right approach, but the logic is not clear to me, and I think it may also confuse other potential readers. This is another area where the manuscript's conciseness comes at a cost of clarity and careful explanation of the underlying logic.

The reviewer is right that we sacrificed clarity for conciseness on this point. One way in which did this is that we skipped too quickly over the initial discussion of regression to the mean in the Introduction. We have now expanded upon this issue in **paragraph 3 of Section 1.1**, in order to set up the logic for our methods from the outset, emphasizing that **"The most effective control is to use independent sub-sets of trials to index performance and to calculate estimation error"**. Further on, in **paragraph 5**, we highlight that we used this approach in our previous study: **"To eliminate regression to the mean, we measured task skill during a preliminary block of trials, independent of our measures of task performance and metacognition."** These changes to **Section 1.1** should make it more obvious to the reader why we go on to use separate sub-sets of trials for the estimation of skill and performance in the present study.

However, the reviewer has identified a problem with our original plan, which we had overlooked. The three sets of puzzles that make up our stimulus set are exactly parallel (i.e. the same set of logical puzzles, implemented with different shapes). Reserving one set for the estimation of task skill, as we had planned to do, will not eliminate regression to the mean, because the performance measure is drawn from puzzle sets containing logically identical puzzles. We have now corrected this shortcoming, and propose instead to estimate skill and performance from independent puzzle sub-sets, administered in separate baseline and test phases, with no puzzles directly parallel between phases (**Methods Section 2.2, paragraph 2**). We are very grateful to the reviewer for pointing out this shortcoming, and we think that the revised design is much stronger.

7. Minor points:

(a) (Please pardon the self-citation) it may be useful to consult Heck & Krueger (2015, JEP:G) for a decision-theoretic analysis of error and bias in the claim to be above average (or not) after completing a single task.

We now cite this paper, in relation to the better-than-average effect (Section 1.1, paragraph 4).

(b) I have two comments on the proposed samples size and power analysis. First, the authors used a corrected alpha level of .01, which seems appropriate, but I did not see an explicit statement that the authors will use this as the criterion for their statistical tests. Second, I suspect that a substantial number of participants will have heard of the DKE before (especially if they are psychology students) – perhaps more than the authors allow for in their power analysis. I appreciate the commitment to analyze the data before and after exclusion, and I also think it may be useful to examine the DKE in those participants who report having knowledge of it.

Following the reviewer's recommendation to take more of an estimation approach, our statistical hypothesis tests are now limited to two (the test of the DKE for relative and for absolute estimation errors), and so our corrected alpha level is now .025. We explicitly state that this is the criterion for our statistical tests (Methods Section 2.6).

We would be happy to propose a separate analysis for those participants who have heard of the DKE before, but we severely doubt that there would be enough data. This is based on our previous study (<https://doi.org/10.1037/xge0000579>), which included 182 participants across two experiments, recruited from the same University sources (not specific to psychology degrees). Only one of these participants, when asked at debrief, said they were familiar with the DKE. Perhaps US students are more knowledgeable than their UK counterparts?

(c) I think I may be missing something in Hypothesis 2: the authors predict a correlation between d' and $(\text{meta-}d'/d')$, but the latter term contains the former, and so they must be correlated. As d' increases, then the fractional term must decrease. Is the hypothesis that the observed correlation will differ from this analytical truth?

Indeed, the prediction is that the observed correlation would be positive, rather than negative. We did not sufficiently emphasise the predicted direction of this correlation in the previous draft, but we now emphasise that this prediction is directionally distinct from the default expectation of a negative correlation (Section 2.7.3, paragraph 3).

(d) For outcome-neutral condition #2, the term “and/or” suggests flexibility in how to proceed if the resulting correlations do not both achieve .3. I share the authors' prediction that these correlations will exceed .3 for absolute and relative EE, but the language should be more explicit about what would happen if one correlation was above .3 and another was below .3.

Our requirements have slightly changed, in that we now require these correlations to be significantly negative (rather than negative and stronger than .3). However, to address the reviewer's point, we now explicitly state that, “Further analyses will be unable to support any

conclusions about the underlying causes of the DKE, for relative or absolute EE, unless the DKE for that type of estimation error is significant.”

Thank you for your work; I look forward to the future of this manuscript.

Thank you very much for your thoughtful and thorough review.

Reviewer: 2

Comments to the Author(s)

This Stage 1 registered report sets out to address a foundational question in metacognition research – the extent to which the celebrated Dunning-Kruger effect (DKE) is a true metacognitive deficit, or not. This is a timely question that is well suited to psychophysical assays of metacognition that do not suffer from the ambiguities in interpretation of the one-shot measures in the classic DKE studies. The study proposal is state-of-the-art in this regard, and I think the authors are well placed to make a definitive contribution to this literature. I mainly have comments about the design and analysis approach that I hope will be helpful as the authors finalise their study plans.

1) My main concern is that the adaptation of the abstract reasoning task adapts it from 4-AFC to a yes/no task – i.e. asking the subject is the solution correct, or not. Moving away from 4-AFC is necessary to allow estimation of meta-d', but I worry that the yes/no version introduces other problems, particularly that response-conditional asymmetries in meta-d' may interact with the hypotheses under test. In both perceptual and memory experiments, it is well-known that metacognitive sensitivity following “no” responses (i.e. responding that a stimulus was absent, or the word was new/not on the list in recognition memory experiments) is substantially worse than metacognitive sensitivity following “yes” responses (Fleming & Dolan, 2010 Consciousness & Cognition; Kanai et al., 2010 Consciousness & Cognition; Higham et al., 2009 JEP:LMC; Meuwese et al. 2014 AP&P). Responding correct/incorrect is formally equivalent to responding yes/no (and the response is framed as Y vs. N), and therefore I worry it will lead to similar metacognitive asymmetries which are orthogonal to the main question of interest. To avoid this why not use a 2AFC task – i.e. the subject needs to respond which out of two options is the correct solution? If there is a strong reason why the yes/no version needs to be used, response-conditional estimation of meta-d' should be employed to allow for these asymmetries in the type 2 ROC fit. If this is not done, then the standard meta-d' model will be unable to accommodate response-conditional divergence between the two type 2 ROC curves and model fits will be poor (https://github.com/metacoglab/HMeta-d/blob/master/Matlab/exampleFit_rc.m for a simulation of this).

We are very grateful to for this advice (and for follow-up advice provided by email). Following this suggestion, we have converted the task to a 2AFC format, presenting a correct and an incorrect solution on each trial. This is now described in **Methods Section 2.2-2.3**.

However, although this increases the value of the data for the extraction of metacognitive measures, it halved the total number of puzzle items available (from 180 to 90). In order to offset this, we have devised a way to boost the available puzzles, creating novel items by

transformation of puzzles from the original set. The transformation steps are briefly described in Methods Section 2.2, and explained in more detail in Supplementary material (S2).

2) A second concern is conceptual. The dual-burden hypothesis is clearly stated in the introduction – that similar psychological resources contribute to both performance and metacognition, such that the worst performers are also the metacognitively least able. But the way this hypothesis relates to the constructs of metacognitive sensitivity (meta-d') and metacognitive efficiency (meta-d'/d') is less clear. On p. 16, it's stated that the dual-burden hypothesis predicts an identity between meta-d' and d'. This is fine, but also psychologically weak, as this relationship would also fall out of a first-order SDT model with no metacognitive “capacity” to speak of. Indeed, this is why the meta-d' framework is useful, because it allows correcting for this influence of performance on measures of metacognition. Instead, a psychologically stronger form of the dual burden hypothesis, and the one that seems most aligned to a metacognitive interpretation of the DKE, is that metacognitive efficiency, corrected for performance, scales with d'. This is discussed at the top of p. 17, but I found it odd that it was not considered a test of the general dual burden idea.

The main purpose of the proposed study is to examine the dual-burden hypothesis within a more precisely-formulated theoretical and computational framework for metacognition. We agree that it is less than perfectly clear how the dual-burden hypothesis relates to the constructs of metacognitive sensitivity and metacognitive efficiency within this model, because the hypothesis was not developed with respect to these constructs, and has only been stated in quite vague verbal terms.

That said, verbal statements of the hypothesis invariably involve some assertion of identity between cognitive and metacognitive processes/resources, as exemplified by the quotes given in response to Reviewer#1's point 1d. If we take these assertions at face value, then they would make the dual-burden hypothesis equivalent to a first-order model, as we argue in Introduction Section 1.3. The empirical prediction is then that metacognitive sensitivity should be strongly related to cognitive sensitivity.

We fully expect to find this result. And we fully agree that, if this is all there is to the dual-burden hypothesis, then it is psychologically weak, because a positive correlation between cognitive and metacognitive sensitivity is practically guaranteed within your model of metacognition, and would not imply any true differences in metacognitive processing between good and poor performers.

Therefore, we also consider a 'stronger' version of the dual-burden hypothesis, which is not necessarily implied by prior statements of that hypothesis, but which you have previously suggested as a possible interpretation of it (Fleming & Lau, 2014). This is that metacognitive efficiency may increase with task ability, such that there are real differences in the quality of metacognitive processing between good and poor performers.

We have made substantial revisions to Introduction Section 1.3, to better distinguish between 'weak' and 'strong' readings of the dual-burden hypothesis, making it clear that the only the former is implied by the prior statements of that hypothesis, and that the latter would be a psychologically stronger, more interesting version of it. We refer back to these ideas in paragraphs 2-3 of Section 2.7.3. We hope that this strikes a balance between being true to

prior statements of the dual-burden hypothesis, but flexible about considering stronger possible versions of the same general idea.

(In passing, we note that the dual-burden hypothesis also requires that cognitive AND metacognitive differences both drive the DKE. This would not seem to be possible under a first-order model, because metacognition could have no predictive power over and above cognitive sensitivity, unless these are at least partially decoupled. The relative importance of cognitive and metacognitive differences in the DKE will be explored by our path analysis.)

Minor points:

The authors might want to consider using the metacognitive threshold measure proposed by Sherman et al. (<https://www.biorxiv.org/content/10.1101/361543v1>) rather than metacognitive bias, as this has more stable psychometric properties and is independent of variation in meta-d'.

We appreciate the suggestion, and we have read about the metacognitive threshold measure with interest. However, we have decided that the more simple metacognitive bias measure is both more easily understandable, and more appropriate, given the relatively limited trial numbers (which may not robustly support the estimation of metacognitive threshold).

p. 13, the HMeta-d toolbox employs Bayesian parameter estimation using MCMC sampling, not maximum likelihood estimation.

Thank you for pointing out this lapse, which we have now corrected in **Methods Section 2.4.2.**

- **Will the HMeta-d toolbox be used in single-subject or group estimation mode? If the latter, note that single-subject parameter estimates will be shrunk to the group mean, which is less useful for subsequent correlation analyses. Instead, a powerful approach is to embed tests of correlations within a hierarchical regression (see https://github.com/metacoglab/HMeta-d/blob/master/Matlab/exampleFit_group_regression.m). But given that the current design has a decent number of trials per subject (180), sticking to single-subject Bayesian estimation may be sufficient and more straightforward.**

We are using the HMeta-d toolbox in single-subject estimation mode. This is now stated in **Methods Section 2.4.2.**

Signed: Steve Fleming

We thank Professor Fleming for his very valuable expert advice. If evaluating the revised manuscript, we hope that he will bring to our attention any errors that we may have made in describing the conceptual or technical aspects of the extraction of metacognitive variables.

Appendix B

Responses to reviewers

Reviewer: 1

Comments to the Author(s)

I found this revision to be extremely responsive to my comments and I no longer have any major concerns. Below, I am noting a few minor blemishes that should be fixed before acceptance. Otherwise, I am grateful to the authors for their thoughtful revisions and clear responses to my review. And I am very much looking forward to seeing the data.

Minor concerns:

In the introduction, two back-to-back paragraphs begin with a "however," clause, which is unsightly.

This is now fixed (we deleted one 'however', and joined two short paragraphs). Many thanks for spotting this and pointing it out.

For the relative and absolute self-estimates, I noticed that one is on a 1-100 scale and the other is on a 0-100 scale. Unless the authors feel that it is theoretically or practically important to keep this difference, I would consider changing one of these so that they are both on the same scale.

We appreciate the desire for symmetry here, but there is a practical reason that the scales differ. Specifically, the absolute estimate is to be compared to absolute performance, which is scored on a 0-100 scale (from chance to perfect); whereas the relative estimate is to be compared to a percentile rank, which is therefore a 1-100 scale. The response scales therefore put the subjective estimates on scales that match the objective scores. We now make it clear in the text that "the scale for the relative response will not register any estimate lower than 1"

There would be other ways to address this problem (for instance, we could ask for a relative estimate from 0-100, and rescale the objective percentile to a similar 101 point scale), but we think that such alternatives are less desirable than the current design.

The acronym "ART" (p. 26) does not seem to be introduced or defined -- could this be a carryover from the original submission?

Yes, this was a hangover from previous version. It has now been fixed. Many thanks for the attention to detail.

I noticed a typo in the link in footnote 7: "souceforge"

Fixed. Ditto.

Reviewer: 2

Comments to the Author(s)

The authors have comprehensively addressed my previous comments, and I look forward to seeing the results of their experiment. Just one minor correction - as JAGS uses precision and not variance, the variance of the prior on meta-d' in the single-subject model is 2, not 0.5.

Fixed. Many thanks for spotting this.

Appendix C

Psychology
SCHOOL of PHILOSOPHY, PSYCHOLOGY and LANGUAGE SCIENCES

The University of Edinburgh
7 George Square
Edinburgh EH8 9JZ

Telephone +44 (0) 131 650 3440

Fax +44 (0) 131 650 3444

Email r.d.mcintosh@ed.ac.uk

08 September 2022

Dear Prof. Chambers (Chris),

We are very pleased to submit this Stage 2 Registered Report, and we are grateful for your flexibility around the fact that data collection was stalled for nearly 2 years by the pandemic. (IPA 08/04/2020; start of testing 28/01/2022).

We now submit the full Stage 2 manuscript, with updated Supplementary material. The tracked version shows changes within the Introduction and Methods with respect to the Stage 1 version, including agreed changes to the protocol, along with dates of approval. Please note that we have also updated the manuscript title, to better reflect the main focus of the experiment.

The full data and analysis code are archived at <https://osf.io/u8kt4/>. This is also the URL to the archived Stage 1 IPA manuscript, as stated on the title page of the manuscript (page 1). The archive includes “SSK_participants.csv” file, which is the laboratory log giving the date of testing for every participant. No data (other than pilot data) were collected prior to IPA. We have also archived the stimulus materials/experiment control programs.

We look forward to receiving your evaluation of our completed Registered Report.

Yours, on behalf of the authors,

Robert McIntosh

Professor of Experimental Neuropsychology, University of Edinburgh, **Europe**

Appendix D

Psychology
SCHOOL of PHILOSOPHY, PSYCHOLOGY and LANGUAGE SCIENCES

The University of Edinburgh
7 George Square
Edinburgh EH8 9JZ

Telephone +44 (0) 131 650 3440

Fax +44 (0) 131 650 3444

Email r.d.mcintosh@ed.ac.uk

07 November 2022

Dear Prof. Chambers (Chris),

We now submit the final revisions for this Stage 2 Registered Report. We have implemented all the minor changes helpfully suggested by Steve Fleming (and Reviewer#1 did not have any further changes to suggest). These are tracked in the revised manuscript, and we also submit a clean version.

The full data and analysis code are archived at <https://osf.io/u8kt4/>. This is also the URL to the archived Stage 1 IPA manuscript, as stated on the title page of the manuscript (page 1). The archive includes “SSK_participants.csv” file, which is the laboratory log giving the date of testing for every participant. No data (other than pilot data) were collected prior to IPA. We have also archived the stimulus materials/experiment control programs.

Many thanks for your patient handling of this Registered Report.

Yours, on behalf of the authors,

Robert McIntosh

Professor of Experimental Neuropsychology, University of Edinburgh, **Europe**